# Mechanism of centromere recruitment of the CENP-A chaperone HJURP and its implications for centromere licensing

Dongqing Pan [1], Kai Walstein [1], Annika Take[1], David Bier[1], Nadine Kaiser[2] & Andrea Musacchio[1,3]

Nucleosomes containing the histone H3 variant CENP-A are the epigenetic mark of centromeres, the kinetochore assembly sites required for chromosome segregation. HJURP is the CENP-A chaperone, which associates with Mis18α, Mis18β, and M18BP1 to target centromeres and deposit new CENP-A. How these proteins interact to promote CENP-A deposition remains poorly understood. Here we show that two repeats in human HJURP proposed to be functionally distinct are in fact interchangeable and bind concomitantly to the 4:2:2 Mis18α:Mis18β:M18BP1 complex without dissociating it. HJURP binds CENP-A:H4 dimers, and therefore assembly of CENP-A:H4 tetramers must be performed by two Mis18αβ:M18BP1:HJURP complexes, or by the same complex in consecutive rounds. The Mis18α N-terminal tails blockade two identical HJURP-repeat binding sites near the Mis18αβ C-terminal helices. These were identified by photo-cross-linking experiments and mutated to separate Mis18 from HJURP centromere recruitment. Our results identify molecular underpinnings of eukaryotic chromosome inheritance and shed light on how centromeres license CENP-A deposition.

[1] Department of Mechanistic Cell Biology, Max Planck Institute of Molecular Physiology, Otto-Hahn-Straße 11, 44227 Dortmund, Germany. [2] Department of Chemical Biology, Max Planck Institute of Molecular Physiology, Otto-Hahn-Straße 11, 44227 Dortmund, Germany. [3] Centre for Medical Biotechnology, Faculty of Biology, University Duisburg-Essen, Universitätsstrasse, 45141 Essen, Germany. Correspondence and requests for materials should be addressed to D.P. (email: dongqing.pan@mpi-dortmund.mpg.de) or to A.M. (email: andrea.musacchio@mpi-dortmund.mpg.de)

During cell division in eukaryotes, kinetochores provide a crucial point of microtubule attachment and are essential for chromosome bi-orientation on the mitotic spindle[1]. On each chromosome, the kinetochore emerges from a specialized chromosomal domain named the centromere, whose hallmark is the strong enrichment of centromeric protein A, a histone H3 variant (CENP-A)[2–5]. Despite its very high relative abundance in comparison to other chromosome domains, CENP-A at centromeres co-exists with an excess of canonical H3[6], and yet it is absolutely required for kinetochore assembly. Substantial biochemical and structural evidence has now accumulated that CENP-A becomes embedded in an octameric nucleosome broadly similar to canonical nucleosomes[2,7]. Notable sequence differences, however, are sufficient to allow direct recruitment to the CENP-A nucleosome of two inner kinetochore proteins in the so-called constitutive centromere-associated network (CCAN), CENP-C and CENP-N, in turn permitting the hierarchical assembly of the entire kinetochore[1].

The levels of CENP-A are constant across generations, suggesting that the amount of newly deposited CENP-A matches the levels of existing CENP-A[8–10]. With the exception of a handful of organisms, however, the enrichment of CENP-A within the centromeric domain appears to be independent of specific DNA sequences[2–5]. This observation is the basis of the concept that centromere identity, which exquisitely depends on CENP-A, is inherited epigenetically. This has shifted the focus onto the mechanisms that allow the centromeric levels of CENP-A to be maintained through cell division, leading to the pioneering discovery of factors involved in new CENP-A deposition[11–14].

In many organisms, including humans, dilution of CENP-A occurs during DNA replication, when the CENP-A pool is equally partitioned to the sister chromatids and the resulting vacancies are filled with histone H3[15]. To compensate for replication-coupled dilution, new CENP-A is then deposited in the very early G1 phase of the cell cycle[16,17]. Once deposited, CENP-A is then stably inherited, with little or no dissipation even over extremely long periods[16,18–20]. At least four core factors of the CENP-A loading machinery have been identified. In humans, these include the two-subunit Mis18 complex (Mis18α and Mis18β; relevant proteins are illustrated schematically in Fig. 1a), Mis18-binding protein 1 (M18BP1, also known as KNL2), and Holliday junction recognition protein (HJURP)[11–14,21]. In addition, several accessory factors and regulators of CENP-A deposition have been identified, including, among others, RSF1, MgcRacGAP, Condensin II, and KAT7[22–26].

HJURP is a histone assembly factor that stabilizes soluble CENP-A:H4 dimers before incorporation into centromeric nucleosomes in late telophase/early G1[13,14,27]. The CENP-A:H4 binding region of human (Hs) HJURP, in the protein's N-terminal region, defines a class of CENP-A assembly factors with common ancestry. It includes the *Saccharomyces cerevisiae* protein Scm3, which acts as deposition factor for the CENP-A ortholog Cse4[28–31]. Sequence similarity of these factors is limited to the CENP-A:H4 binding domain[28]. A region in the central domain (CD) of HsHJURP (also identified as mid domain, HMD) has been implicated in DNA binding[32]. Two additional sequence-related HJURP C-terminal domains (HCTD1 and HCTD2) within the carboxy terminal half of the protein[28] are sufficient to promote robust centromere recruitment of HJURP in the G1 phase[33]. Here, we refer to the HCTDs as repeat 1 and repeat 2 (R1 and R2; Fig. 1a, b. An alignment of mammalian HJURP is shown in Supplementary Fig. 1).

Because HJURP localization is sufficient for CENP-A deposition[34–36], several mechanisms control the timing and localization of HJURP recruitment. Central to these mechanisms are M18BP1, Mis18α, and Mis18β. Mis18α and Mis18β form a tight 2-subunit complex (which we refer to here as the Mis18core of the Mis18 complex) and interact tightly but in a regulated, transient manner, with M18BP1 to assemble the Mis18 complex (Fig. 1c). The Mis18 complex precedes HJURP to centromeres and is required for its recruitment there[12,25,34,36–41]. While the mechanism of centromere recruitment of the Mis18 complex remains partly unclear, binding to CENP-A nucleosomes or CENP-C appears to contribute[35–39,42–47]. HJURP itself may specify additional interactions with inner kinetochore proteins[35,40,41,48].

In analogy with the licensing events that limit the initiation of DNA replication to once per cell cycle, several factors have been proposed to promote licensing steps that limit CENP-A deposition to once per cycle[3]. Among the factors required for deposition, positive and negative regulation by the kinase activities of the polo-like kinase 1 (PLK1) and cyclin-dependent kinases 1 and 2 (CDK1/2), respectively, have emerged for their prominence. PLK1 associates with the Mis18 complex at kinetochores in telophase/early G1, and its activity is required for deposition[49]. Cyclin-dependent kinase (Cdk) phosphorylation of HJURP prevents binding to Mis18 and centromere localization[32,50,51] (Fig. 1c, left). Cdk phosphorylation of M18BP1 also inhibits its association with Mis18core and centromere recruitment[49,50,52–54]. As a further licensing step, it has been proposed that binding of HJURP to a Mis18α:Mis18β core tetramer activates HJURP for CENP-A deposition while causing dissociation of the Mis18α: Mis18β tetramer into a dimer that is unable to rebind centromeres[8]. Subsequent work, however, re-examined the stoichiometry of the Mis18 core complex and found it to consist of a 4:2 hexamer[52,54], as discussed more thoroughly in the Results section. Finally, M18BP1 release from the centromere was shown to be required for efficient CENP-A deposition[50]. Collectively, these events have been interpreted as manifestations of a global licensing mechanism controlling the deposition machinery so that deposition is limited to a single round.

Work of biochemical reconstitution revealed aspects of kinetochore organization that led us to hypothesize a new mechanistic model for why newly deposited CENP-A is fixed on the levels of existing CENP-A[1]. Specifically, various observations suggest that the substrate recognized by the CENP-A deposition machinery is an inner kinetochore structure consisting of neighboring CENP-A and H3 nucleosomes[1] connected by inner kinetochore proteins in the CCAN, especially CENP-C and CENP-T[20,55–58] (Fig. 1c, right). We speculated that processing of this substrate through eviction of the paired H3 nucleosome and its replacement with a new CENP-A nucleosome will cause the deposition machinery to dissociate in a manner similar to the dissociation of enzymes from their products[1]. While this model remains unproven, it provides a conceptual alternative to models postulating that prevention of multiple rounds of deposition requires the active dissociation of the Mis18core complex by HJURP or the active dissociation of M18BP1 from centromeres[8,50]. Both the dinucleosome model and the two dissociation models discussed above predict that CENP-A content will be doubled during deposition, but identify different causes for this phenomenon: exhaustion of the reaction substrate on one hand, and active dissociation of the enzyme on the other.

Testing the significance of these various models requires a detailed understanding of, and exquisite control over, the interactions that promote CENP-A deposition. While progress has been made in this direction, remaining gaps of knowledge prevent formal proof of competing hypotheses. Here, we contribute to fill this gap by analyzing the mechanism of the interaction of HJURP with Mis18core and M18BP1 using a powerful combination of tools in vitro and in vivo. With results that seem irreconcilable with the Mis18 dissociation model[8], we demonstrate that the

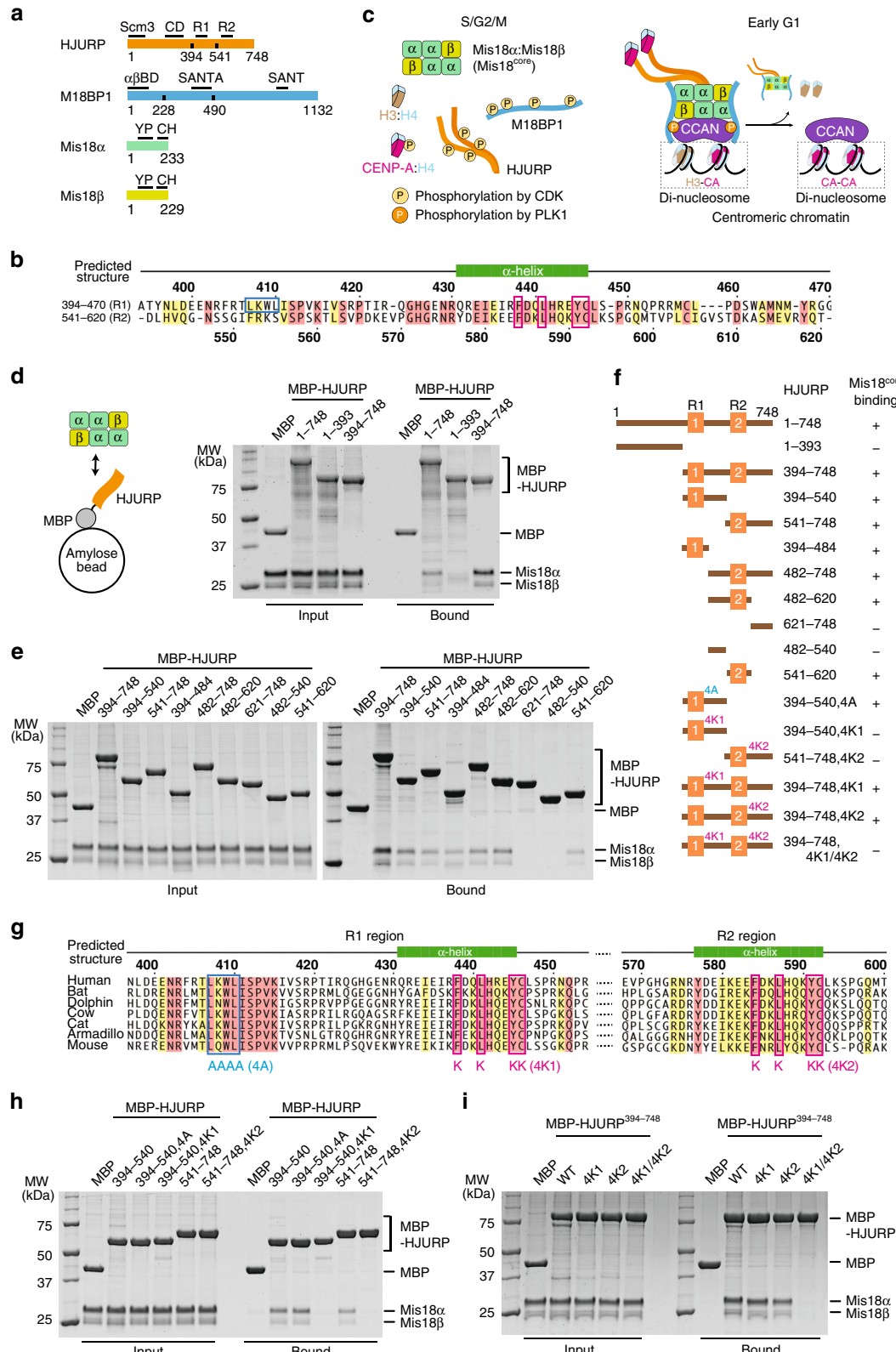

interaction with HJURP does not cause dissociation of the Mis18$^{core}$ complex, regardless of the presence of M18BP1. We show that the CENP-A deposition complex contains M18BP1, Mis18$^{core}$, and HJURP in defined stoichiometries, with the latter present in a single copy. The isolated R1 and R2 repeats are monomeric, and largely functionally interchangeable if provided with the appropriate dosage. They bind, as part of the same HJURP molecule, to equivalent sites on the two trimeric C-terminal helical bundles of the Mis18 complex. Conversely, we find no evidence that HJURP dimerization is important for the interaction with Mis18, as proposed previously[33]. By amber-codon suppression[59,60] we map the Mis18-binding sites for the R1 and R2 repeat by mass spectrometry, and validate them by mutational analysis in vitro and in cells. Collectively, our results

**Fig. 1** Both HJURP C-terminal repeats, R1 and R2, bind Mis18$^{core}$ complex. **a** Domain organization of HJURP, M18BP1, Mis18α, and Mis18β. CD central domain, R1 repeat 1, R2 repeat 2, αβBD Mis18α:Mis18β binding domain, YP Yippee domain, CH C-terminal helix. **b** Sequence alignment of R1 and R2 of HsHJURP. Multiple sequence alignment was performed using ClustalW[85]. Secondary structure prediction was performed using PSIPRED[86]. Residues identical in all sequences are shaded red, and conserved substitutions with similar properties are shaded yellow. Blue and magenta boxes indicate residues mutated to 4A, 4K1, or 4K2 as indicated in **g**. **c** Schematic model of the phosphorylation-regulated assembly of the CENP-A deposition machinery. **d**, **e** Amylose-resin pull-down assays were performed to identify the Mis18$^{core}$-binding regions of HJURP. Proteins were incubated at 5 μM in binding buffer A containing 30 mM HEPES pH 7.5, 300 mM sodium chloride, 1 mM TCEP, 0.01% Tween-20. The gels were stained with Coomassie brilliant blue (CBB). **f** Summary of the pull-down results. **g** Sequence alignment of HJURP sequences around R1 and R2 regions from different mammalian species. **h**, **i** Amylose-resin pull-down assays for testing the binding of Mis18$^{core}$ and HJURP variants with 4A, 4K1, and 4K2 mutations

unveil fundamental aspects of a crucial molecular mechanism required for chromosome segregation and intergenerational genetic inheritance, and pave the way for future mechanistic studies.

## Results

**Interaction of R1 and R2 in HJURP with Mis18$^{core}$.** The C-terminal region of HJURP is sufficient to bind the Mis18$^{core}$ complex[33,41,51]. Within this region, the R1 repeat promotes binding to the Mis18$^{core}$ complex, whereas R2 promotes HJURP dimerization[33,51]. Secondary structure prediction programs indicate that the C-terminal region of HJURP is, for the most part, intrinsically disordered (Supplementary Fig. 1), with few exceptions, like the R1 and R2 repeats, which are predicted to contain a single α-helix. Given the generally modest stability of isolated α-helices, we surmise that this conformation is only adopted after binding to the Mis18 complex (R1) or after the proposed dimerization (R2).

To validate the interaction mechanism, we generated several C-terminal fusions of HJURP fragments to maltose binding protein (MBP) and immobilized them on solid phase. We then incubated the fusion proteins with the Mis18$^{core}$ complex, washed away unbound proteins, and evaluated the composition of beads by SDS-PAGE analysis. In line with the previous reports[33,51], full-length HJURP (HJURP$^{1-748}$) and HJURP$^{394-748}$ bound to Mis18$^{core}$, while HJURP$^{1-393}$ did not (Fig. 1d). With the same assay, we dissected the determinants of the HJURP:Mis18$^{core}$ interaction, confirming that R1 (HJURP$^{394-484}$) is sufficient to bind Mis18$^{core}$ (Fig. 1e. A summary is in Fig. 1f).

Unexpectedly, HJURP fragments encompassing R2 (e.g. HJURP$^{541-620}$) also bound Mis18$^{core}$, whereas fragments encompassing the segment between R1 and R2 (HJURP$^{482-540}$) or the region C-terminal to R2 (HJURP$^{621-748}$) did not (Fig. 1e, f). Binding of R2 of HJURP to the Mis18$^{core}$ complex contradicts a previous report[8]. Because a shorter R2 construct (HJURP$^{555-748}$ instead of HJURP$^{541-748}$ used here) had been used in the previous study[8], we compared the Mis18$^{core}$ binding proficiency of HJURP$^{555-748}$ and HJURP$^{541-748}$. HJURP$^{541-748}$ bound Mis18$^{core}$ markedly better than MBP-HJURP$^{555-748}$ (Supplementary Fig. 2a), likely explaining the different outcomes of the binding experiments. In conclusion, R1 and R2 of HJURP bind the Mis18$^{core}$ complex also in isolation and without requiring other segments of HJURP.

Sequence motifs encompassing residues 438–446 of R1 and 584-592 of R2 are highly conserved (Fig. 1g), and we tested their contribution to Mis18$^{core}$ binding. Four-lysine mutants of repeat 1 (4K1) or repeat 2 (4K2) abrogated the interaction with Mis18$^{core}$ (Fig. 1h). While neither the 4K1 nor the 4K2 mutant disrupted binding of HJURP$^{394-748}$ (which contains both repeats) to Mis18$^{core}$, combining mutations abolished the interaction (Fig. 1i). Conversely, a four-alanine mutation (4A) replacing conserved residues 407–410 in HJURP did not affect its binding to Mis18$^{core}$ (Fig. 1g, h). Collectively, this shows that at least in this binding assay,

at relatively high concentrations of interacting species, R1 and R2 are interchangeable towards Mis18$^{core}$ complex binding.

R2 has been proposed to promote and be sufficient for HJURP dimerization[33]. To verify this claim, we applied sedimentation velocity analytical ultracentrifugation (AUC), a method of choice for accurate determination of molecular mass. In our analysis, we identified MBP-HJURP$^{541-748}$ (encompassing R2) as a monomer at 5 μM concentration, with no evidence of dimeric species (Supplementary Fig. 2b). A modest level of dimerization (or possibly aggregation) was only observed with the HJURP$^{394-540}$ construct (encompassing R1), the bulk of which, however, was monomeric too, as previously observed[33]. We conclude that R1 and R2, at the relatively high concentration used in these assays, can bind independently to the Mis18$^{core}$ complex and that they are, respectively, largely or exclusively monomeric in solution in isolation from the rest of the HJURP sequence.

**Correct dosage of R1 or R2 is required for CENP-A deposition.** Because both R1 and R2 of HJURP support Mis18$^{core}$ complex binding, we were curious to assess if they can individually support CENP-A deposition. For this, we adapted a previously described strategy based on selective fluorescent labeling of newly deposited SNAP-tagged CENP-A in HeLa cells[16,52] (Fig. 2a). Stable HeLa cell lines for tetracycline-inducible co-expression of EGFP-NLS (nuclear localization signal) or EGFP-HJURP with SNAP-tagged CENP-A (CENP-A$^{SNAP}$) were generated by transfection of Flp-In T-REx HeLa cells[61] with pcDNA5/FRT/TO derived plasmids. CENP-A$^{SNAP}$ expression was adjusted by mutating internal ribosome entry site in the pcDNA5/FRT/TO plasmids. Protein expression was confirmed by western blotting (Supplementary Fig. 3a). After irreversible labeling of newly deposited CENP-A$^{SNAP}$ with a fluorescent dye targeting the SNAP-tag (SNAP-Cell 647-SiR), we readily visualized a strong centromere signal in early G1 phase HeLa cells. In line with the essential role of HJURP in CENP-A deposition, depletion of HJURP by RNAi (Supplementary Fig. 3b) abrogated CENP-A loading (Fig. 2b).

We used this assay to test the ability of individual HJURP transgenes, expressed from the tetracycline-inducible promoter, to functionally complement the deleterious effects from depleting endogenous HJURP. EGFP-HJURP$^{WT}$ (wild type) strongly labeled inner kinetochores and its expression in HJURP-depleted cells resulted in almost complete recovery of CENP-A$^{SNAP}$ deposition at inner kinetochores (Fig. 2c). The introduction of the individual 4K1 mutation in R1 or of the 4K2 mutation in R2, or their combination in a single construct, prevented centromere localization of HJURP and failed to support new CENP-A$^{SNAP}$ incorporation at significant levels in cells depleted of endogenous HJURP (Fig. 2c–e). All three mutant constructs had strong dominant-negative effects on the ability of cells retaining wild-type HJURP (i.e., treated only with transfection reagent rather than anti-HJURP siRNA) to promote new CENP-A deposition (–siRNA, Fig. 2c–e), possibly because they compete with endogenous HJURP for CENP-A or other interaction partners. Thus, although both R1 and R2 of HJURP bind the

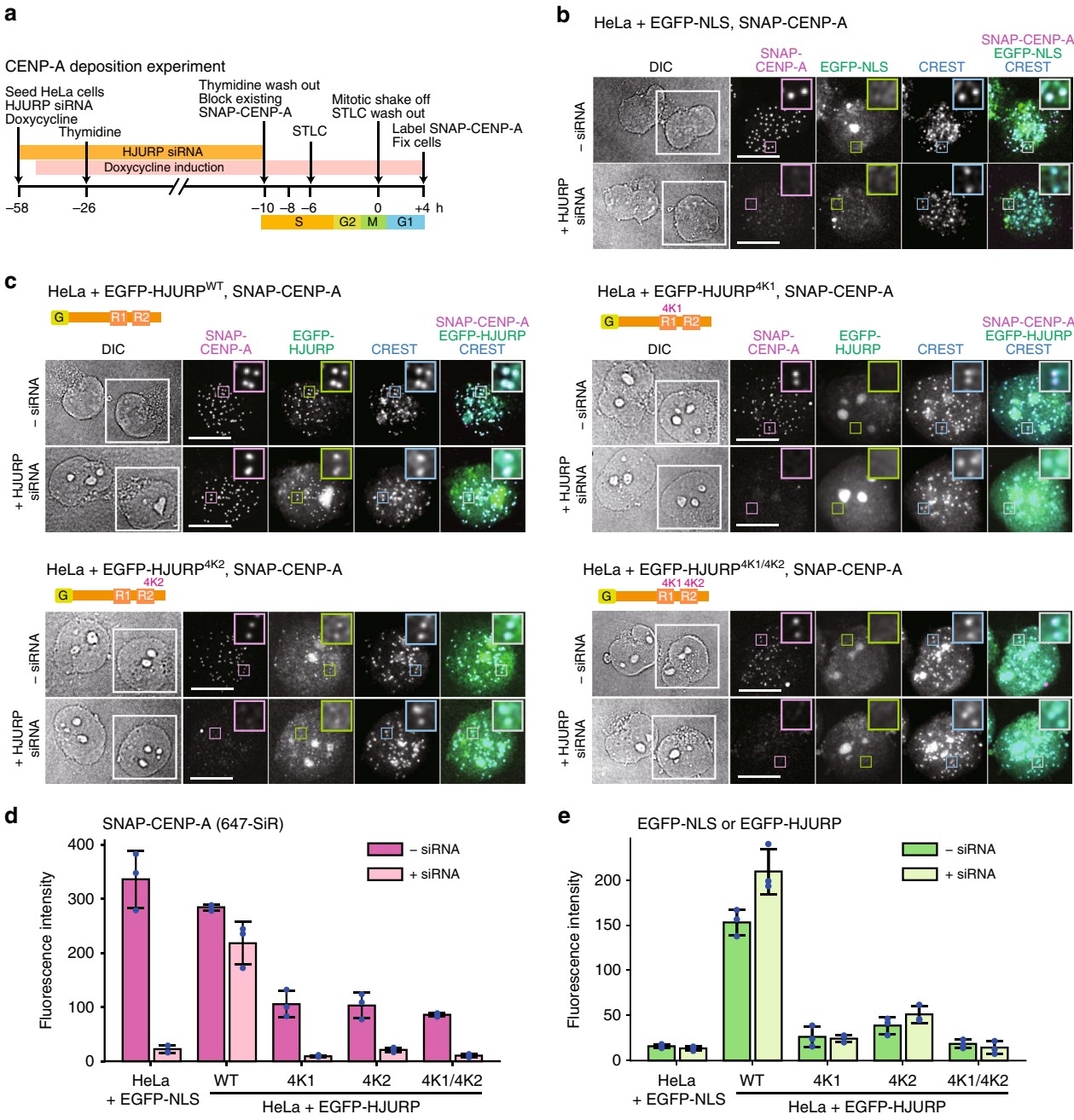

**Fig. 2** Four-lysine mutations in HJURP R1 or R2 disturb HJURP localization and CENP-A deposition. **a** Schematic of the CENP-A deposition experiment for testing functionality of HJURP variants. SNAP-Cell 647-SiR was used to label newly produced SNAP-CENP-A at the time point +4 h. **b, c** Representative images showing SNAP-CENP-A fluorescence and EGFP-NLS or EGFP-HJURP variants in fixed HeLa cells treated as described in panel **a**. Centromeres were visualized with CREST sera. Control cells were treated with transfection reagent (Lipofectamine RNAiMAX) in the absence of HJURP siRNA. DIC, differential interference contrast. One side of the white square in the DIC panel represents 20 μm. White scale bars indicate 10 μm. All cell biological experiments in this paper were repeated at least three times. **d** Quantification of the centromere fluorescence intensity of SNAP-CENP-A. Centromere spots were detected using the images of CREST channel and were applied to the images of other data channels. In each experiment, a mean value of centromere fluorescence intensity was obtained from at least 340 centromere spots from at least 20 early G1 cells. The highest 10% and the lowest 10% intensity values were considered outliers and excluded. The bar graph represents mean values from the three replicate experiments (blue dots indicate the mean values from each experiment). Error bars indicate standard deviations. The ways of quantification and representation described here apply to other figures displaying experiments in HeLa cells. **e** Quantification of the centromere fluorescence intensity of EGFP signal. Source data are provided as a Source Data file

Mis18[core] complex in a biochemical assay, they are both required in a cellular assay probing HJURP centromere recruitment and CENP-A deposition.

The strong defect on CENP-A deposition observed with individual mutations in the R1 or R2 motifs may reflect functional specialization, or, alternatively, reduced dosage of functionally equivalent regions. The in vitro characterization in Fig. 1 supports the second hypothesis, but we sought formal proof by asking if CENP-A deposition was rescued by HJURP constructs harboring tandem copies of the same repeat. HJURP constructs where R1 or

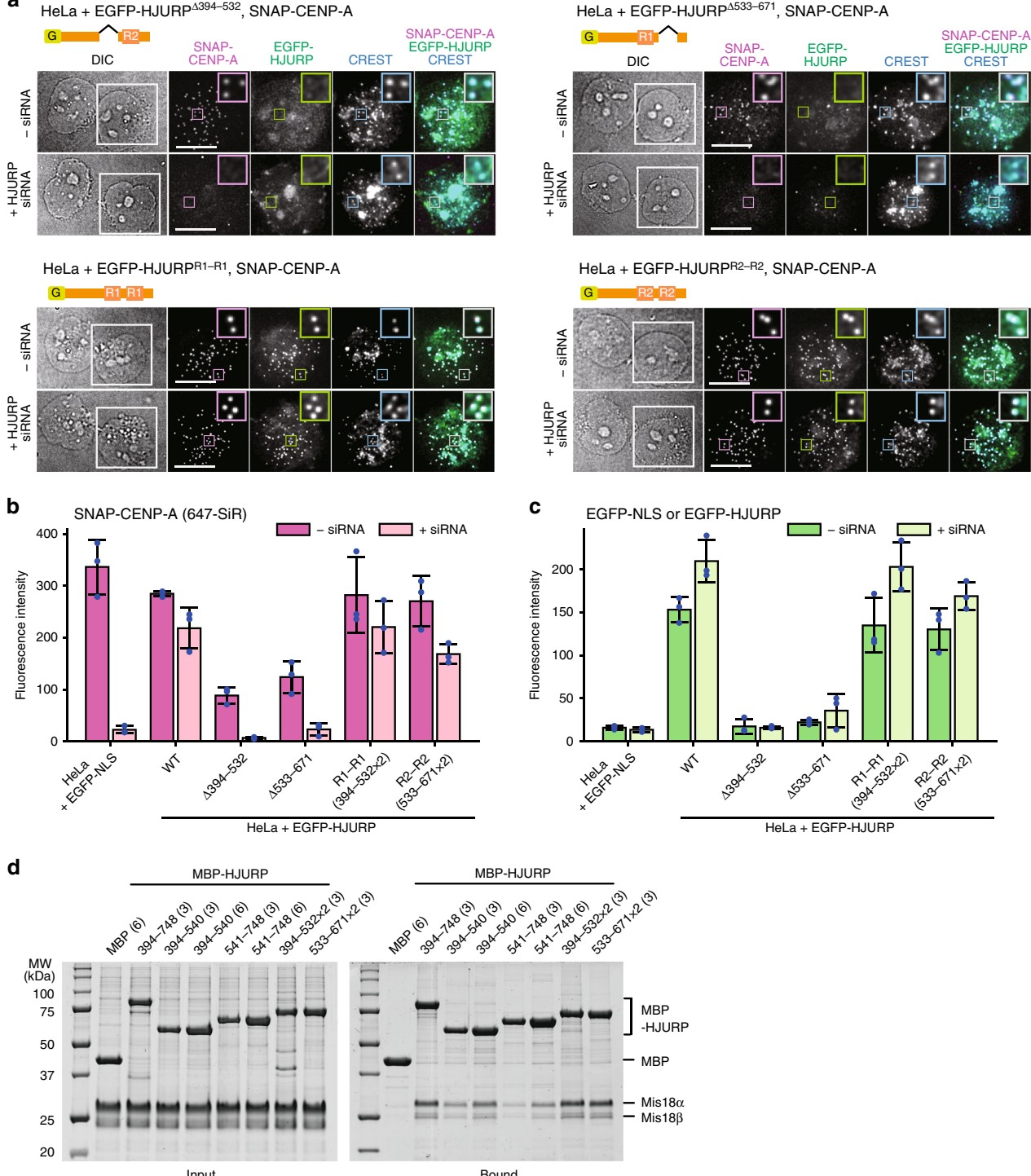

**Fig. 3** Tandem repeats of HJURP R1 or R2 rescue the localization of HJURP and CENP-A deposition. **a** Representative images showing the fluorescence of SNAP-CENP-A and EGFP-HJURP variants in fixed HeLa cells treated as described in Fig. 2a. White scale bars indicate 10 μm. **b**, **c** Quantification of the centromere fluorescence intensity of SNAP-CENP-A or EGFP-HJURP variants. The bar graphs represent mean values from three replicate experiments (blue dots indicate the mean values from each experiment). Error bars indicate standard deviations. **d** SDS-PAGE results of amylose-resin pull-down assays showing the increase of HJURP binding to Mis18core when tandem repeats of R1 and R2 were used. The numbers in the parenthesis indicate the concentrations (μM) of the proteins used in the assays. Source data are provided as a Source Data file

R2 had been deleted (HJURP$^{\Delta 394–532}$ or HJURP$^{\Delta 533–671}$, respectively) failed to localize to kinetochores, were unable to load new CENP-A$^{SNAP}$ onto centromeres, and resulted in a strong dominant-negative effect on CENP-A deposition in presence of endogenous HJURP (Fig. 3a–c), precisely as already observed with

the 4 K mutants. In contrast, constructs harboring identical tandem repeats (HJURP$^{R1-R1}$ or HJURP$^{R2-R2}$) localized robustly to kinetochores and rescued CENP-A deposition to levels similar to those achieved by expression of HJURP$^{WT}$. In pull-down assays with recombinant proteins in vitro, HJURP constructs consisting of

tandem R1 or R2 repeats showed increased affinity for Mis18[core] than constructs with single R1 or R2 repeats (Fig. 3d). Collectively, these results indicate that R1 and R2 are at least largely functionally redundant, and that their appropriate dosage is crucial. A single HJURP repeat is sufficient for the interaction with Mis18[core] at the relatively high concentrations of the in vitro assays, but insufficient for kinetochore recruitment and CENP-A loading at the lower cellular concentrations, making the presence of two repeats necessary, largely regardless of their identity. In combination with the demonstration that both R1 and R2 are monomeric, our results do not support a role of R2 dimerization in HJURP function[33].

**The Mis18α N-terminal region modulates HJURP binding.** Next, we tried to identify molecular determinants of the interaction of Mis18[core] with HJURP. We have shown previously that the Mis18[core] complex binds tightly to residues 1–60 of M18BP1 (M18BP1[1–60]), and less tightly to residues 61–140 (M18BP1[61–140])[52]. In agreement with the previous results, Mis18[core] bound immobilized MBP-M18BP1[1–60] tightly in solid phase binding assays (Fig. 4a). Mis18[core] also bound MBP-M18BP1[61–140], but apparently with lower binding affinity, and at levels that were comparable to those obtained with immobilized single-repeat baits MBP-HJURP[394–540] and MBP-HJURP[541–748] (Fig. 4a).

The Mis18[core] complex was originally characterized as an oligomer with a proposed 2:2 stoichiometry of Mis18α and Mis18β[8,62], but additional work demonstrated a 4:2 Mis18α:Mis18β hexamer[52,54]. Crucial to the establishment of this stoichiometry are the assembly of a 3-helical bundle of the C-terminal helices and the dimerization of the Yippee domains[52,54,62]. The 4:2 Mis18[core] hexamer interacts with the N-terminal segments of two protomers of M18BP1, generating a 4:2:2 assembly[22,39,52,54].

Deletion of the predicted C-terminal helices (CH, Fig. 1a) affects the oligomerization state of Mis18[core], resulting in simple Mis18α:Mis18β dimers, held together by the Yippee domains, which remain able to bind M18BP1[52,54]. On the contrary, a CH-deleted Mis18[core] (Mis18α:[1–191]Mis18β[1–189]) decreased or even abrogated binding to all four immobilized MBP bait constructs (Fig. 4b), indicating that structural integrity of the complex or the CH regions themselves (or both) are required for the interaction (as discussed below).

In an unanticipated outcome of these experiments, we observed that deletion of the N-terminal regions of Mis18α and Mis18β (Mis18α:[78–233]Mis18β[65–229]) did not affect its hexameric state (Supplementary Fig. 4a) but strongly promoted binding to MBP-HJURP[394–540] and MBP-HJURP[541–748], to levels that were as robust as those observed with M18BP1[1–60] (Fig. 4c). This effect was recapitulated by deleting the N-terminal tail of the Mis18α subunit (Mis18α[78–233]:Mis18β[1–229], Fig. 4d), but not the N-terminal tail of the Mis18β subunit (Mis18α[1–233]:Mis18β[65–229]) (Fig. 4e). The CH regions of Mis18α and Mis18β were necessary for HJURP binding also when combined with deletions of the N-terminal regions (Mis18α[78–191]:Mis18β[73–189]) (Fig. 4f). Of note, this effect of Mis18α[78–233]:Mis18β[65–229] was specific for the R1 or R2 regions of HJURP, because binding to MBP-M18BP1[1–60] or MBP-M18BP1[61–140] remained strong or weak, respectively, even with the longest deletions tested (Fig. 4c–e), confirming that the N-terminal regions of the Mis18 complex subunits are not strictly necessary for M18BP1 binding[52].

A strongly conserved motif between residues 26 and 54, also predicted to adopt α-helical conformation, is present in the N-terminal region of mammalian Mis18α sequences (Fig. 4g, Supplementary Fig. 4b). We asked if this region of Mis18α contributes to the observed modulation of the binding affinity for

HJURP. To test this, we created various deletions of the Mis18α N-terminal region that either excluded or included the predicted α-helix within the conserved motif. The deletion mutants Mis18α[26–233]:Mis18β[1–229] and Mis18α[36–233]:Mis18β[1–229], which preserve the predicted α-helix, did not bind the MBP control but bound the HJURP single-repeat baits MBP-HJURP[394–540] and MBP-HJURP[541–748] with the relatively low apparent affinity already observed with full-length Mis18[core] (Fig. 4h–j). Conversely, additional deletions that removed the predicted α-helix (Mis18α[56–233]:Mis18β[1–229] and Mis18α[78–233]:Mis18β[1–229]) strongly increased the apparent binding affinity for the HJURP repeats (Fig. 4h–j). The same N-terminal deletions of the Mis18 complex did not show progressively increased binding affinity for the M18BP1[61–140] construct (Fig. 4k; in fact, the apparent binding affinity appeared to decrease slightly with longer deletions, revealing a marginal effect in binding) but were equally good binders of the M18BP1[1–60] construct (Supplementary Fig. 4c). We conclude that the effects of the N-terminal deletions are HJURP specific.

The results of these binding assays, which are collectively summarized in Fig. 4l, predict that the Mis18α N-terminal tail might be dispensable for centromere recruitment of HJURP and new CENP-A deposition. To test this, we depleted Mis18α by RNAi interference (Supplementary Fig. 5a, b). As expected, the Mis18 complex was necessary for CENP-A[SNAP] deposition and its depletion prevented new CENP-A[SNAP] deposition onto centromeres in early G1 (Supplementary Fig. 5c). This deficiency was rescued by expression of Mis18α[WT] or Mis18α[56–233] transgenes (Supplementary Fig. 5b, c, quantified in panel D), indicating that the N-terminal region of Mis18α is not required for the loading reaction. In fact, Mis18α[56–233] decorated centromeres even more brightly than Mis18α[WT], and complemented more effectively the depletion of endogenous Mis18α, in agreement with the results in vitro.

**HJURP and Mis18 subunits form a stoichiometric complex.** Binding of HJURP has been shown to cause dissociation of a Mis18[core] tetramer into dimers and release of its subunits from centromeres[8]. Because in our previous studies we characterized the Mis18α:Mis18β assembly for containing a hexameric Mis18-core and for being very stable in vitro[52,54], we wanted to verify whether HJURP caused dissociation of the Mis18[core] complex. For this, we performed size-exclusion chromatography (SEC) analyses, where proteins are separated based on their size and shape. As expected, MBP-HJURP[394–748], which encompasses both the R1 and R2 regions, bound full-length Mis18[core] (Fig. 5a). Shorter HJURP segments encompassing only the R1 or R2 regions (MBP-HJURP[394–540], or MBP-HJURP[541–748]), however, showed weaker binding to Mis18[core] (Fig. 5b, c), in agreement with the solid phase binding assays in Fig. 1. Nevertheless, the same HJURP constructs bound strongly to the Mis18α[56–233]:Mis18β[1–229] HJURP-super-binder complex (Fig. 5d–f). There was no evidence that the interaction of HJURP causes dissociation of the Mis18[core] complex as previously reported[8].

To identify the copy number of HJURP binding to the Mis18[core] complex, we purified Mis18α[56–233]:Mis18β[1–229]:HJURP[394–748] and Mis18α[56–233]:Mis18β[1–229]:MBP-HJURP[394–748] complexes by SEC and analyzed them by sedimentation velocity AUC (Fig. 5g, h, Supplementary Fig. 6a–c). The Mis18α[56–233]:Mis18β[1–229] HJURP-super-binder retains the typical 4:2 stoichiometry of the full-length Mis18[core] complex[52,54], indicating that deletion of the Mis18α N-terminal region in the HJURP-super-binder does not affect the stoichiometric ratios of the complex. The observed molecular weights of Mis18α[56–233]:Mis18β[1–229]:HJURP[394–748] and Mis18α[56–233]:Mis18β[1–229]:MBP-HJURP[394–748] complexes indicated

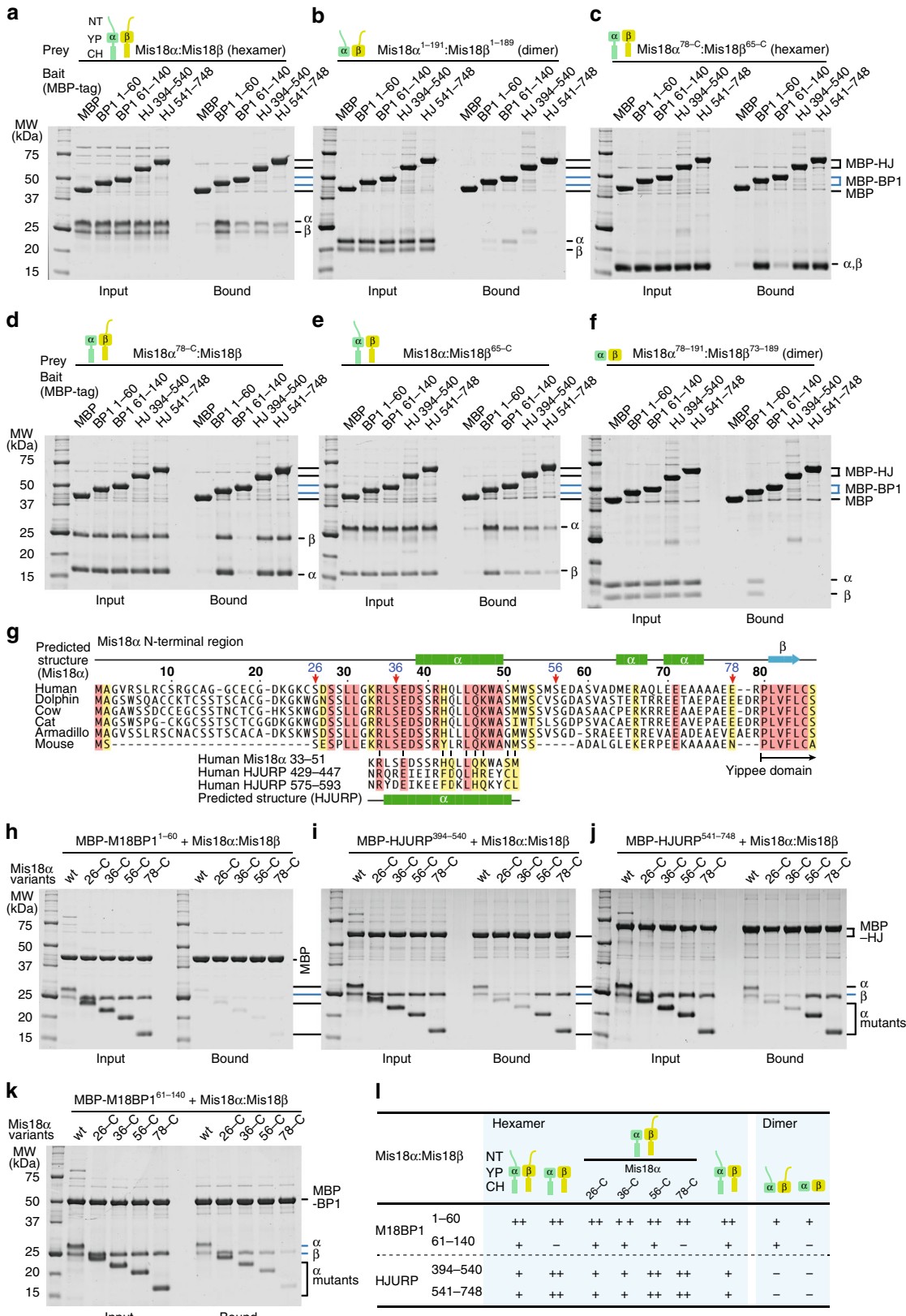

binding of only one copy of HJURP$^{394–748}$ to Mis18α$^{56–233}$:Mis18β$^{1–229}$ complex (Fig. 5g, h, Supplementary Fig. 6a–c).

Next, we asked if full-length Mis18$^{core}$ or Mis18α$^{56–233}$:Mis18β$^{1–229}$, when combined with MBP-M18BP1$^{1–228}$ and MBP-HJURP$^{394–748}$, gave rise to complexes comprising all four subunit types. MBP fusion constructs encompassing different

segments of the bi-partite Mis18$^{core}$-binding site in the M18BP1 N-terminal region (MBP-M18BP1$^{1–140}$, MBP-M18BP1$^{1–60}$, and MBP-M18BP1$^{61–140}$) were incubated with Mis18$^{core}$ or Mis18α$^{56–233}$:Mis18β$^{1–229}$. We then asked if addition of HJURP$^{394–748}$ (containing the R1 and R2 regions of HJURP) disrupted this complex. To increase the solubility of

**Fig. 4** Systematic interaction analysis reveals binding mechanism of HJURP on Mis18core. **a–f** Amylose-resin pull-down assays were performed using MBP-tagged M18BP1[1–60], M18BP1[61–140], HJURP[394–540], and HJURP[541–748] with different constructs of Mis18core. Binding buffer B containing 15 mM HEPES pH 7.5, 150 mM NaCl, 1 mM TCEP, and 0.01% Tween-20 was used for the pull-down assays presented in panel **b** and **f**. Binding buffer A was used for other pull-down assays. **g** Amino-acid sequence alignment showing the N-terminal region of mammalian Mis18α. The sequence similarity among highly conserved regions in HJURP R1, R2 and the N-terminus of Mis18α is shown. Regions predicted to form α-helices are indicated by green bars. **h–k** Amylose-resin pull-down assays were performed to identify the residues of Mis18α that modulate the binding of HJURP to Mis18core. **l** A table summarizing the pull-down results presented here and in Supplementary Fig. 4c. single plus and double plus indicate weak and strong interactions, respectively

HJURP[394–748], we fused it to a mutant MBP that had been rendered unable to bind amylose (defined as MBPKR, and containing the two previously reported mutations E111K and W230R[63]). MBPKR-HJURP[394–748] did not bind to the MBP control or the MBP-M18BP1 baits in the absence of Mis18core, but it bound robustly when Mis18core was supplemented, indicating that HJURP and M18BP1 bind concomitantly to the Mis18 complex (Fig. 6a, b). We corroborated this conclusion with SEC analyses. MBP-M18BP1[1–228] readily bound full-length Mis18core or the Mis18α[56–233]:Mis18β[1–229] construct as we reported previsouly[52], but did not bind MBP-HJURP[394–748] (Fig. 6c), indicating that M18BP1 and HJURP do not interact directly. When the Mis18core was added, a complex containing all four subunits was observed, without evidence of Mis18core dissociation (Fig. 6d, e), thus demonstrating that M18BP1, Mis18α, Mis18β, and HJURP can form a single stable complex.

As a further demonstration of this result, we resorted again to sedimentation velocity AUC. The complex of the Mis18α[56–233]:Mis18β[1–229] HJURP-super-binder with FAMMBP-M18BP1[1–228] shows the familiar 4:2:2 stoichiometry[52,54] (Fig. 6f, g, Supplementary Fig. 6d, e). Next, we added two or four equivalents of MBP-HJURP[394–748] to the 4:2:2 Mis18α[56–233]:Mis18β[1–229]:MBP-FAMM18BP1[1–228] assembly and assessed the molecular mass of the resulting complex. This unequivocally showed that only one equivalent of MBP-HJURP[394–748] is incorporated in the 4:2:2 complex, giving rise to a 4:2:2:1 stoichiometry, without dissociation of Mis18α from Mis18β (Fig. 6f, g, Supplementary Fig. 6d, e).

**HJURP-binding sites on Mis18.** In a previous study, the C-terminal regions of Mis18α and Mis18β were proposed to be sufficient to recruit HJURP to an ectopic chromosome locus and for an interaction in vitro with 1:1:1 stoichiometry[8]. We re-examined these previous findings in light of our new observations that both the R1 and the R2 repeat of HJURP interact with the Mis18core, and that the CH regions form trimers rather than dimers[52,54]. We used amber-codon suppression methodology[64] to introduce, in different positions of the R1 or R2 regions of HJURP, unnatural amino-acids with photo-activatable cross-linker groups, p-benzoyl-L-phenylalanine (Bpa) and 3'-azibutyl-N-carbamoyl-lysine (AbK) (Fig. 7a). MBP-HJURP[394–540] or MBP-HJURP[541–748] containing either Bpa or AbK at 12 different positions (indicated in Fig. 7b) of the R1 or R2 segments were combined with Mis18α[56–233]:Mis18β[1–229], and the cross-linking (XL) reaction was activated with 365 nm ultraviolet (UV) light on the resulting complex (shown schematically Fig. 7c, left). Prominent high-molecular weight bands appeared after UV exposure, indicating that MBP-HJURP constructs containing Bpa or AbK had reacted with Mis18α and Mis18β (Supplementary Fig. 7a). Cross-linked products were further purified using Ni[2+] affinity resin, washed with urea buffer, and subjected to proteolysis and mass spectrometry (MS) analysis. We identified many cross-links of the Bpa- or AbK-incorporated HJURP fragments with Mis18α or Mis18β (Fig. 7d–f). The vast majority of residues targeted by Bpa or AbK mapped to the predicted 3-helical bundle encompassing the CH regions of Mis18α or Mis18β, in line with

previous findings implicating these regions in HJURP binding[8]. Introduction of the cross-linkers in the R1 or the R2 segments resulted in similar patterns of cross-links.

These results indicate that R1 and R2 each bind one of the two predicted CH trimers in the Mis18core complex (Fig. 7c, right), a finding that is consistent with the functional equivalence of the R1 and R2 fragments demonstrated above. A parsimonious expectation based on the 4:2 stoichiometry and mode of inter-subunit interaction is that the Mis18core complex is two-fold symmetric. Thus, we expect the interactions of R1 and R2 with the CH domains to be semi-equivalent. In Fig. 4, we showed that the CH domains are necessary for Mis18core to bind HJURP. To assess if the CH domains are sufficient to bind HJURP, we tested binding of immobilized HJURP[394–748] (which covers the R1 and R2 repeats, Fig. 1f) to trimeric constructs only encompassing the CH domains of Mis18αβ[52]. To mimic the stoichiometry of CH domains in the Mis18core complex, we fused GST in frame with Mis18β[175–229] to cause dimerization of the trimers[65,66]. Nevertheless, no binding of the Mis18 CH regions to immobilized HJURP[394–748] was observed (Supplementary Fig. 7b), indicating that in addition to the CH domains, contacts with the rest of the Mis18 complex increase the overall binding affinity for HJURP.

**A separation of function mutant prevents HJURP recruitment.** We used program CCBuilder 2.0[67] to model the interaction of the CH helices in their 2:1 Mis18α:Mis18β stoichiometry and thus visualize their 3-dimensional organization. At this moment, there is no published result defining the orientation of α-helices in the CH trimer. Judging from the mapping of the cross-links obtained, the binding site for HJURP R1 and R2 is formed by one CH helix of Mis18α and one of Mis18β in a parallel configuration (Fig. 7e). The model also suggests that the third predicted helix of Mis18α does not affect the HJURP-binding pocket. We modeled it in a parallel configuration with the other two helices, but recognize that it might assemble in an anti-parallel configuration without consequences for our conclusions. Residues that were targeted by Bpa and AbK in the R1 or R2 repeats had very similar footprints on this model (Fig. 7f, g).

The emerging surface formed by the parallel CH helices of Mis18α and Mis18β contains various conserved residues, including Val211Mis18α, Ala241Mis18α, and His207Mis18β (Fig. 8a). We hypothesized therefore that this region is implicated in the interaction with R1 and R2. To test this, we created individual point mutations at Val211Mis18α, Ala214Mis18α, and His207Mis18β. Met214Mis18β was also tested because it was cross-linked to both F584BpaHJURP and Y591BpaHJURP. The V211DMis18α mutant was almost completely impaired in its binding to MBP-HJURP[394–748] (Fig. 8b). Another mutant, A214DMis18α, was also partly impaired, whereas two additional mutants H207AMis18β and M214DMis18β did not seem to alter the affinity for HJURP. Importantly, the penetrant V211DMis18α mutant appeared to retain all its binding affinity for M18BP1 (Fig. 8c). Thus, V211DMis18α is a separation-of-function mutant that selectively impairs the interaction of the Mis18 complex with HJURP but does not impair the interaction of the Mis18 complex with M18BP1. Importantly, the V211DMis18α mutation did not

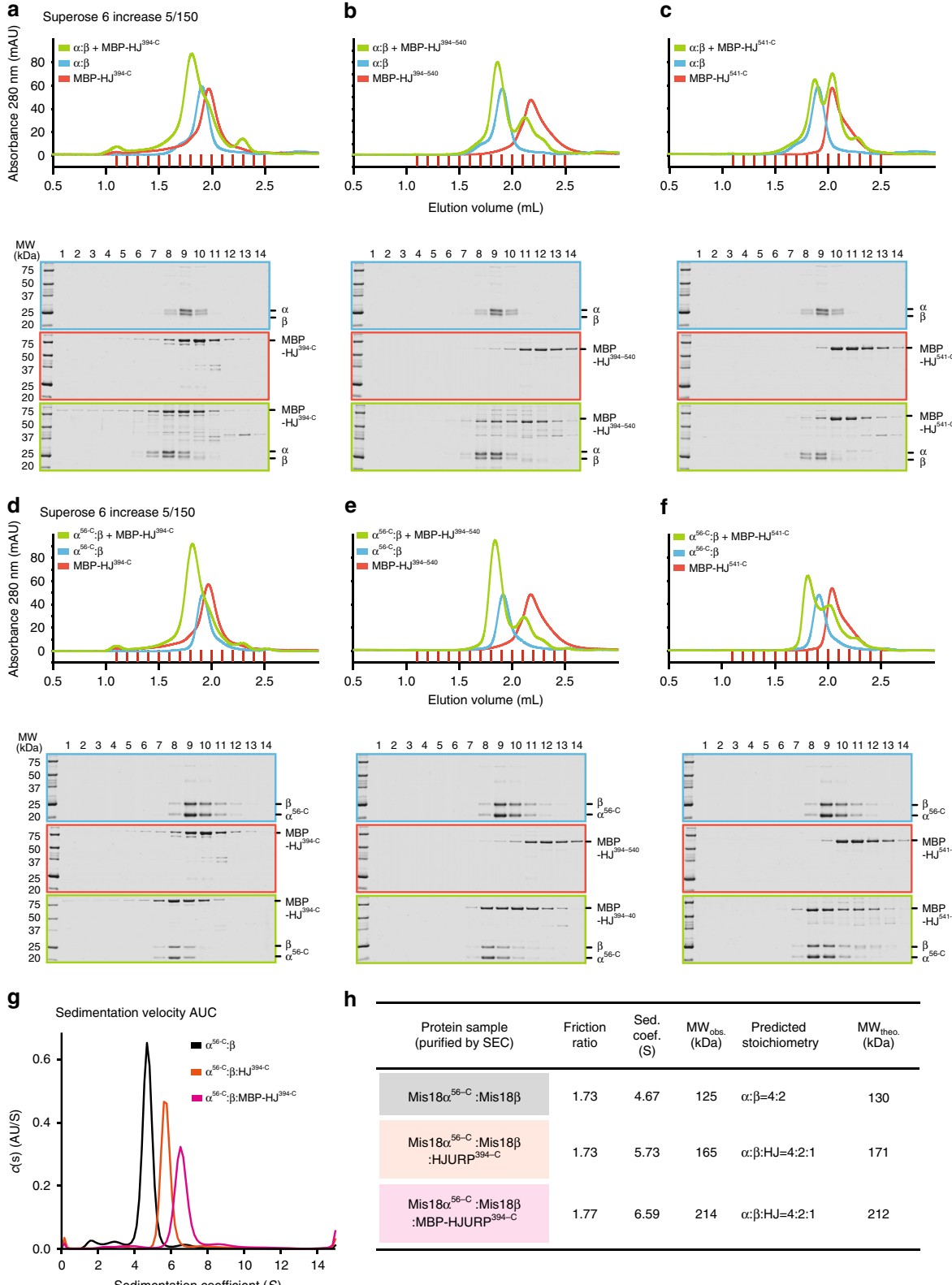

**Fig. 5** Mis18$^{core}$ forms stable complex with only one copy of HJURP. **a–f** Analytical SEC experiments were performed to confirm that HJURP fragments bind full-length Mis18$^{core}$ or Mis18α$^{56-C}$:Mis18β without dissociation of these complexes. SDS-PAGE gels were stained with CBB. **g** Sedimentation coefficient distributions obtained from the sedimentation velocity AUC experiments using the purified samples described in Supplementary Fig. 6a, b. Data profiles used for curve-fitting analyses are shown in Supplementary Fig. 6c. **h** A table showing the values obtained from the AUC experiments in panel **g**. Sed. coef. sedimentation coefficient, MW$_{obs.}$ observed molecular weight, MW$_{theo.}$ theoretical molecular weight

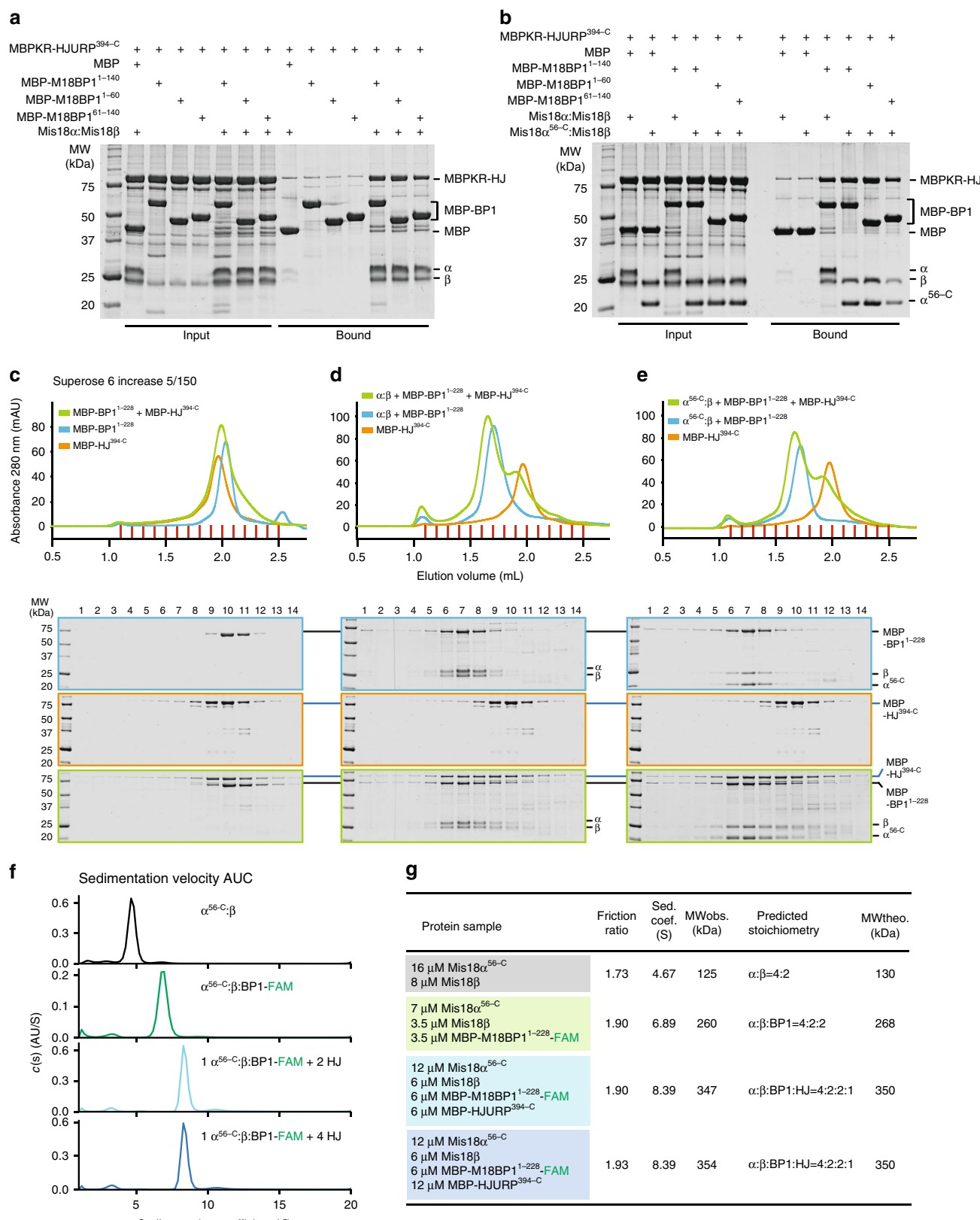

impair the stoichiometry of the Mis18core complex, thus ruling out that impaired HJURP binding was caused by an indirect effect on the stability of the Mis18 complex (Supplementary Fig. 7c).

To validate these results in vivo, we created stable HeLa cell lines and co-expressed EGFP-Mis18α[V211D] with mCherry-HJURP in cells depleted of endogenous Mis18α (Fig. 8d).

EGFP-Mis18α[V211D] localized to centromeres in the early G1 phase and showed equivalent fluorescence intensity to EGFP-Mis18α[WT]. However, EGFP-Mis18α[V211D] did not rescue the localization of mCherry-HJURP when endogenous Mis18α was depleted (Fig. 8e–g). Furthermore, expression of EGFP-Mis18α[V211D] in presence of endogenous Mis18α had a strong

**Fig. 6** Reconstitution of Mis18-HJURP complex and determination of its molecular stoichiometry. **a**, **b** Results of amylose-resin pull-down assays showing that HJURP forms a complex with M18BP1 only in presence of Mis18$^{core}$. Binding buffer B was used for these pull-down assays. **c–e** Analytical SEC experiments were performed to confirm that both HJURP and M18BP1 bind Mis18$^{core}$ simultaneously without dissociation of Mis18$^{core}$. Proteins were mixed in buffer at the following concentrations: Mis18$^{core}$ (hexamer), 5 μM; MBP-HJURP$^{394-C}$, 10 μM; MBP-M18BP1$^{1-228}$, 10 μM. SDS-PAGE gels were stained with CBB. **f** Sedimentation coefficient distributions obtained from the sedimentation velocity AUC experiments using the protein samples described in the figure. Data profiles used for curve-fitting analyses are shown in Supplementary Fig. 6e. **g** A table showing the values obtained from the AUC experiments in panel **f**. Sed. coef. sedimentation coefficient, MW$_{obs.}$ observed molecular weight, MW$_{theo.}$ theoretical molecular weight. The AUC results of Mis18α$^{56-C}$:Mis18β presented here (**f, g**) are identical with those presented in Fig. 5g, h

dominant-negative effect on centromere recruitment of mCherry-HJURP, likely because it binds endogenous M18BP1 (Fig. 8e–g).

## Discussion

CENP-A is the epigenetic marker of centromeres, and dissecting the molecular basis of its deposition and maintenance through subsequent cell divisions is expected to clarify the foundations of centromere inheritance. A successful molecular model of CENP-A deposition ought to include answers to several crucial questions, including: (1) why is the deposition reaction limited to the existing CENP-A domain? (2) Does the deposition reaction target for eviction and replacement a specific neighboring nucleosome, e.g., an H3 nucleosome marked by particular interactions with neighboring CENP-A nucleosome(s)? And (3) what limits the deposition reaction so that the levels of newly deposited CENP-A match the levels of already existing CENP-A, thus ensuring that the amounts of CENP-A are maintained through the generations?

In recent years, initial answers to these questions have emerged. We have learned that CENP-A deposition occurs during the late telophase/early G1 phase of the cell cycle, when Cdk activity is lowest[16,17]. The deposition reaction is complex, and there is growing evidence that it "reads" particular structures at the centromere to force deposition of new CENP-A very near the existing pool[10,20,35–41,43–46,48,55–58]. In a very interesting recent twist, HJURP has also been implicated in the retention of CENP-A at centromeres during DNA replication, when CENP-A is equally distributed to the sister chromatids[68], thus expanding the functional horizon of the deposition machinery.

The study presented here provides answers that are especially relevant to questions 1 and 3. Our characterization of the interaction between the Mis18$^{core}$ and HJURP forces a revision of several previous claims on its molecular mechanism and significance. HJURP and related proteins only bind to a CENP-A:H4 dimer, not a tetramer[69–71]. Thus, a model postulating HJURP dimerization on its centromere-targeting factors is appealing, as it suggests a mechanism for how a CENP-A:H4 tetramer may be reconstituted[33]. Albeit attractive, this model seems incompatible with our new observations. While we could not raise evidence supporting HJURP dimerization, we provide strong evidence that a single molecule of HJURP binds to the Mis18 complex. Our data are in principle compatible with the possibility that the same Mis18:HJURP complex generates a tetramer in two subsequent steps (with intervening new loading of CENP-A:H4) or that two HJURP:CENP-A:H4 complexes bind to the Mis18 receptor sequentially and deposit two CENP-A:H4 dimers for tetramerization (Fig. 8h, right). Alternatively, and more likely, two closely positioned Mis18:HJURP complexes deposit distinct CENP-A:H4 dimers for tetramerization into the same nucleosome (Fig. 8h, left), a possibility supported by 2:1 stoichiometry of the CCAN:CENP-A nucleosome complex[57].

In their work, Zasadzinska et al. (2013)[33] also reported that non-overlapping HJURP constructs containing either R1 or R2 were both capable of reaching kinetochores, and explained this with a direct kinetochore recruitment function of R1, and a dimerization of R2 that allows it to dimerize with the endogenous

HJURP. We show here instead that deletion of R1 or R2 from full-length HJURP impairs kinetochore recruitment both in presence and absence of endogenous HJURP, and that constructs containing two copies of R1 or R2 localize to kinetochore after depletion of the endogenous protein. A significant technical difference is that we used stable transgene integration at a single chromosomal site, whereas Zasadinska et al.[33] operated under condition of transient-transfection of a pIC113 plasmid containing a constitutive CMV promoter, and thus likely under conditions of over-expression.

Another important conclusion of our study is that binding to HJURP$^{394-C}$ does not change the oligomerization state of the Mis18$^{core}$ complex as previously proposed[8]. Although other regions of HJURP might be responsible for Mis18$^{core}$ dissociation in living cells, the fact that HJURP$^{1-393}$ does not bind Mis18$^{core}$ conflicts with this idea. Furthermore, Nardi et al.[8] observed dissociation of Mis18$^{core}$ by HJURP in vitro when using a C-terminal segment of HJURP with similar span to the one we have used. While we cannot pinpoint the reasons for these differences, we note that our conclusion that only one HJURP$^{394-C}$ binds to the Mis18 octamer and that the complex does not dissociate were obtained with analytical ultracentrifugation, the gold standard for this purpose.

Genetic code expansion coupled with amber-codon suppression has emerged as a method of choice for the introduction of new functionalities in proteins[64]. Here, we used this powerful methodology with two different photo-activatable cross-linkers, coupled with mass spectrometry, to identify the binding site of HJURP on its centromere receptor. We find that the two HJURP repeats R1 and R2 bind on two equivalent 3-helical bundles assembled by the C-terminal regions of two Mis18α and one Mis18β subunits. We also discovered that the N-terminal region of Mis18α contains a sequence motif whose deletion strongly facilitates the interaction of the Mis18$^{core}$ complex with HJURP. This motif aligns with the R1 and R2 repeats (Fig. 4g), leading us to speculate that in the absence of HJURP, it folds intramolecularly on the HJURP-binding sites to occlude them and avoid untimely binding. Our future studies will aim to test this speculative idea and to assess its significance for CENP-A deposition.

Collectively, our findings are particularly relevant to question 3 above, namely what ensures that the levels of newly deposited CENP-A match the levels of existing CENP-A. An attractive idea is that there is a licensing step analogous to the one limiting initiation of DNA replication from any given origin to once per cell cycle[72]. Our observations do not discard the idea altogether, but are inconsistent with the identification of Mis18$^{core}$ dissociation as a termination step of the loading reaction[8]. A conceptually alternative mechanism for question 3 is that the deposition reaction is enzymatic and terminates when all the "substrate", such as the CENP-A-H3 di-nucleosome that we have postulated (see Introduction and ref. [1]), is processed and turned over into a CENP-A-CENP-A di-nucleosome product, causing the release of the "enzyme". This model postulates that centromere licensing of CENP-A deposition consists in the identification of a particular H3 nucleosome introduced during DNA

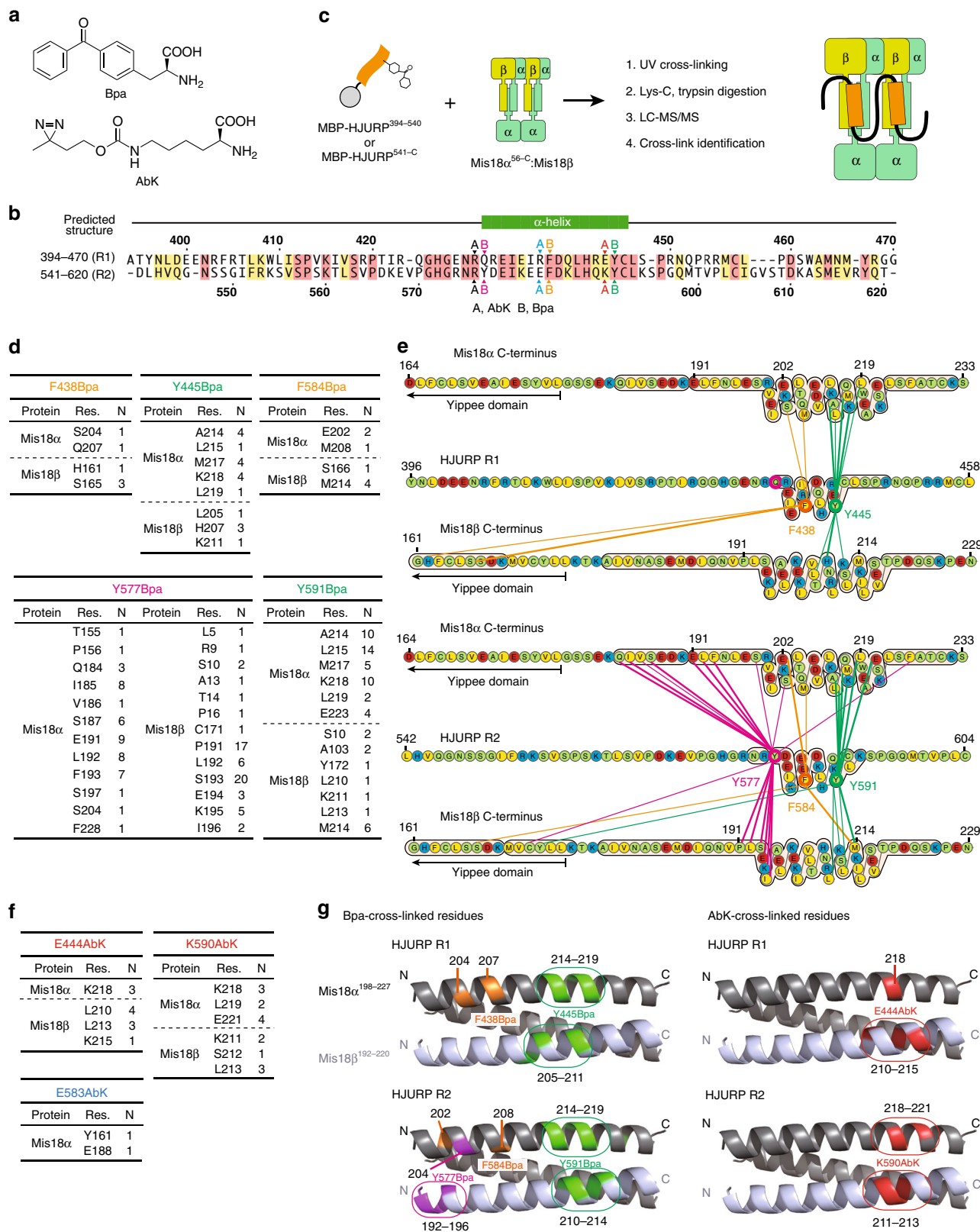

replication and residing in the vicinity of CENP-A, i.e., in the generation of a suitable substrate. We speculate that generation of the substrate entails that this H3 nucleosome readily binds CCAN subunits, including CENP-C and possibly CENP-T, that later attract the deposition machinery. A formal analysis of this model will require the identification of the exact determinants of Mis18 complex recruitment and activation at centromere, i.e., of the exact substrate of the deposition reaction.

## Methods

**Plasmids.** Codon-optimized cDNAs of human Mis18α, Mis18β, M18BP1, and HJURP were purchased from GeneArt. Subcloning and plasmid construction were

**Fig. 7** UV-cross-linking and MS analysis reveal equivalent binding sites of HJURP R1 and R2 on Mis18 C-terminal helix-bundle. **a** Chemical structures of *p*-benzoyl-L-phenylalanine (Bpa) and 3′-azibutyl-*N*-carbamoyl-lysine (AbK). **b** Amino-acid sequences of HJURP showing positions where Bpa or AbK were introduced by amber-codon suppression. Bpa was introduced into position 431, 438, 445, 577, 584, or 591. AbK was introduced into position 430, 437, 444, 576, 583, or 590. **c** Schematic description of the UV-cross-linking experiment followed by MS analysis. **d** Cross-links identified by MS analysis of Bpa-cross-linked samples. Res. residue, N numbers of identification. **e** Schematic representation of cross-links identified between the incorporated Bpa residues in HJURP R1 or R2 and the residues in C-terminal regions of Mis18α and Mis18β. **f** Cross-links identified by MS analysis of AbK-cross-linked samples. **g** Bpa- and AbK- cross-linked residues are highlighted in colors on a 3D model of three-helix-bundle of Mis18α and Mis18β. The details of cross-links identified can be found in Supplementary Data 2

performed using standard restriction-enzyme-based cloning, Gibson cloning[73], and PCR-based site-directed mutagenesis methods. The coding DNA sequences (CDS) of genes of interest in all expression plasmids were verified by DNA sequencing before using for further experiments. A list of plasmids used in this study can be found in Supplementary Data 1.

*Escherichia coli* (*E. coli*) expression plasmids pETDuet-MBP-HJURP-8His variants and pETDuet-MBP-M18BP1-8His variants were generated by inserting PCR-amplified CDSs of HJURP and M18BP1 into pETDuet-MBP-8His vector[52] between the N-terminal MBP-tag and the C-terminal 8His-tag using *Bam*HI and *Xho*I sites. *E. coli* expression plasmids pETDuet-6His-Mis18α-MBP-Mis18β variants were generated from pETDuet-6His-Mis18α-MBP-Mis18β[52] by replacing the CDSs of full-length Mis18 and/or full-length Mis18β with the CDSs of different shorter fragments. Point mutations were introduced by PCR-based site-directed mutagenesis.

Mammalian expression plasmid pcDNA5-EGFP-NLS-IRESv12-SNAP-CENP-A was generated from a modified pcDNA5/FRT/TO plasmid (Thermo Fisher Scientific), pcDNA5/FRT/TO-EGFP-IRES[74]. A CDS of nuclear localization signal (NLS) from simian virus 40 was placed between the CDSs of EGFP and IRES, and the CDSs of SNAP-tag and human CENP-A were subcloned from pETDuet-CENP-A-SNAP-HA-PGK-NeoR[52] using Gibson cloning. Mutations were introduced into IRES according to a previous study[75] to obtain the IRESv12 with reduced translation efficiency. Plasmid pcDNA5-EGFP-NLS-IRESv12-SNAP-CENP-A enables co-expression of EGFP-NLS under the regulation of CMV/TetO₂ promoter and SNAP-CENP-A under the control of IRESv12 with much reduced expression level. The plasmid pcDNA5-EGFP-HJURP-IRESv12-SNAP-CENP-A for co-expression of EGFP-HJURP and SNAP-CENP-A was generated by replacing the CDS of NLS of pcDNA5-EGFP-NLS-IRESv12-SNAP-CENP-A with human HJURP original CDS from genomic DNA (the internal *Bam*HI site was removed) using *Bam*HI and *Xho*I sites. The plasmids for co-expression of EGFP-HJURP variants with SNAP-CENP-A were generated by PCR-based site-directed mutagenesis or restriction-site-based ligation. Especially, pcDNA5-EGFP-HJURP (R1-R1)-IRESv12-SNAP-CENPA was generated by replacing the CDS of NLS of pcDNA5-EGFP-NLS-IRESv12-SNAP-CENP-A with a ligation product of three HJURP DNA fragments that encode HJURP[1–532], HJURP[394–532], and HJURP[672–748]. And pcDNA5-EGFP-HJURP(R2-R2)-IRESv12-SNAP-CENPA was generated by replacing the CDS of NLS of pcDNA5-EGFP-NLS-IRESv12-SNAP-CENP-A with a ligation product of three HJURP DNA fragments that encode HJURP[1–393], HJURP[533–671], and HJURP[533–748]. Plasmids pcDNA5-EGFP-Mis18α-IRESv12-SNAP-CENPA and pcDNA5-EGFP-Mis18α(56-C)-IRESv12-SNAP-CENPA were generated by replacing the CDS of NLS of pcDNA5-EGFP-NLS-IRESv12-SNAP-CENP-A with the codon-optimized CDSs of Mis18α or Mis18α[56–233]. Plasmid pcDNA5-EGFP-Mis18α-P2AT2A-mCherry-HJURP-IRESv12-SNAP-CENP-A for co-expression of EGFP-Mis18α, mCherry-HJURP, and SNAP-CENP-A was constructed by inserting the CDSs of Mis18α, P2AT2A[52] and mCherry-tag between the CDSs of EGFP and HJURP of pcDNA5-EGFP-HJURP-IRESv12-SNAP-CENPA using Gibson cloning. A point mutation of Mis18α V211D was introduced into this plasmid by PCR-based site-directed mutagenesis to obtain pcDNA5-EGFP-Mis18(V211D)-P2AT2A-mCherry-HJURP-IRESv12-SNAP-CENPA.

Plasmid pEVOL-ABK (Addgene #126035) was generated by replacing the CDSs of the amber suppressor tRNA/aminoacyl-tRNA synthetase pairs for Bpa (tRNA$^{Tyr}$/Bpa-RS) of pEVOL-pBpF (Addgene #31190)[76] with the CDSs of the amber suppressor tRNA/aminoacyl-tRNA synthetase pairs for AbK (tRNA$^{Pyl}$/AbK-RS). The CDSs of AbK-RS was subcloned from pDULE-ABK (Addgene #49086)[59] into two positions in pEVOL plasmid. One CDS of AbK-RS was placed between *araBAD* promoter and *rrnB* terminator, and the other CDS of AbK-RS was placed between *glnS'* promoter and *glnS* T terminator according to the original design of pEVOL plasmid. The CDS of tRNA$^{Pyl}$ is placed between *proK* promoter and *proK* terminator. A previously reported mutation (U25G)[77] was introduced to the sequence of tRNA$^{Pyl}$ to improve the efficiency of amber suppression.

### Chemical synthesis of 3′-azibutyl-*N*-carbamoyl-lysine.
Unnatural amino-acid 3′-azibutyl-*N*-carbamoyl-lysine (AbK) **1** (Fig. 7a, Supplementary Fig. 8) was synthesized following the scheme published previously[78].

First, we synthesized diazirine **2** (Supplementary Fig. 8) from 4-hydroxy-2-butanone. Under slow stirring, 4-hydroxy-2-butanone (20 g, 0.22 mol) was added to liquid NH₃ (120 mL) at −78 °C, followed by another 5 h stirring at −78 °C.

Subsequently, hydroxylamine O-sulfonic acid (28.2 g, 0.24 mol) in methanol (200 mL) was added, the reaction mixture was allowed to warm up to the ambient temperature and stirred overnight. The white precipitated was removed by filtration and the volume of the filtrate was reduced to 200 mL. The remaining mixture was cooled to 0 °C, followed by the addition of methanol (200 mL) and freshly distilled triethylamine (30 mL). Subsequently, iodine (~28 g) was added until a persistent iodine coloring remained. The reaction mixture was warmed up to the ambient temperature and stirred for another 2 h. Under reduced pressure the volume of the reaction mixture was reduced to 200 mL. The reaction mixture was diluted with brine (200 mL) and extracted three times with ether (100 mL). Subsequently, the combined organic phases were dried over anhydrous MgSO₄ and the solvent was removed under reduced pressure. Finally, the desired product **2** was obtained in 32% yield (7.2 g) as brown oil.

Next, we activate the diazirine **2** with disuccinimide carbonate. Under argon atmosphere, *N*,*N*'-disuccinimidyl carbonate (6 g, 23.42 mmol) and freshly distilled triethylamine (8 mL) were added to a mixture of the diazirine **2** (1.5 g, 14.98 mmol) in dry acetonitrile (40 mL). The reaction mixture was stirred for 20 h at the ambient temperature and subsequently dried under reduced pressure. The crude product was purified by column chromatography on silica gel using a mixture of acetone and chloroform (20:1). The desired product **3** (Supplementary Fig. 8) was obtained in 97% yield as an orange solid. 1H NMR (400 MHz, CDCl₃) δ = 4.28 (t, J = 6.5 Hz, 2H), 2.75 (s, 4H), 1.77 (t, J = 6.5 Hz, 2H), 1.10 (s, 3H).

Under argon atmosphere, Boc-Lys-OH (4.5 g, 18.28 mmol) was added to a solution of NHS-carbonate (3 g, 12.44 mmol) in dry DMF (50 mL). The reaction mixture was stirred for 35 h at room temperature, poured into water (250 mL) and stirred for another 20 min. Following an extraction with ether (3 × 75 mL), the combined organic phases were washed with brine (2 × 75 mL) and dried over anhydrous MgSO₄. Subsequently, the solvent was removed under reduced pressure and the desired product **4** (Supplementary Fig. 8) was obtained in 79% yield as orange oil. ¹H NMR (400 MHz, Chloroform-*d*) δ 6.14 (s, 0.5H), 5.24 (s, 1H), 4.94 (s, 0.5H), 4.29 (s, 1H), 4.12–3.94 (m, 2H), 3.26–3.13 (m, 2H), 1.92–1.33 (m, 17H), 1.05 (s, 3H).

Finally, the Boc group was removed from **4**. The Boc-amine **4** (1.5 g, 4.03 mmol) and triethylsilane (1.28 mL) were dissolved at the ambient temperature by slow addition of 5% TFA in dichloroethane (65 mL). The reaction mixture was stirred for 18 h and the volatile components were removed under reduced pressure. The remaining residue was dissolved in methanol (24 mL) and precipitated by drop-wise addition into ether (200 mL) under vigorous stirring. The precipitation process was repeated once more and provided the desired product **1** (0.79 g) as a white crystalline solid in 72% yield. 1H NMR (500 MHz, D₂O) δ 3.84 (t, J = 6.0 Hz, 2H), 3.62 (t, J = 9.1 Hz, 1H), 2.97 (t, 2H), 1.81–1.65 (m, 2H), 1.57–1.44 (m, 2H), 1.39–1.34 (m, 2H), 1.28–1.19 (m, 2H), 0.86 (s, 3H). ¹³C NMR (126 MHz, D₂O) δ 174.04, 158.38, 60.40, 54.09, 39.77, 33.13, 29.78, 28.39, 24.89, 21.43, 18.58. LRMS: calc. [M + H]⁺ 273.31, found 273.04.

### Protein expression and purification.
Insect-cell-expressed Mis18α:Mis18β complex used in the pull-down assays was expressed and purified using the method reported previously[52]. Other proteins used in this study, except Bpa-incorporated and AbK-incorporated MBP-HJURP variants, were expressed in *E. coli* cells of BL21-CodonPlus(DE3)-RIL strain (Agilent Technologies, #230240) and purified using the method reported previously[52] with minor modification. *E. coli* cells transformed with expression plasmids were cultured in 2 × YT media (16 g L⁻¹ tryptone, 10 g L⁻¹ yeast extract, 5 g L⁻¹ NaCl) supplemented with ampicillin and chloramphenicol at 37 °C. Protein expression was induced by adding IPTG to the final concentration of 0.2 mM when OD₆₀₀ of the culture reached 0.6 and further incubation at 20 °C for 16 h. Then *E. coli* cells expressing proteins of interest were suspended in buffer HST300 (30 mM HEPES pH 7.5, 300 mM sodium chloride, 1 mM TCEP) containing 1 mM PMSF and 10 mM imidazole and lysed by sonication. The clear supernatant obtained after centrifugation was incubated with Ni resin (Roche) for ~16 h. Protein-bound resin was washed with 100 column volumes of buffer HST300 containing 10 mM imidazole. Proteins were eluted with buffer HST300 containing 400 mM imidazole and concentrated in buffer HST300. The concentration of imidazole was reduced to less than 40 mM by repetitive dilution of the concentrated protein sample with buffer HST300. MBP-HJURP variants were further purified using a Superdex 200 10/300 GL SEC column (GE Healthcare) with buffer HST300. After the Ni-affinity purification, the variants of *E. coli* expressed Mis18α:Mis18β complex were incubated with TEV protease for ~16 h at

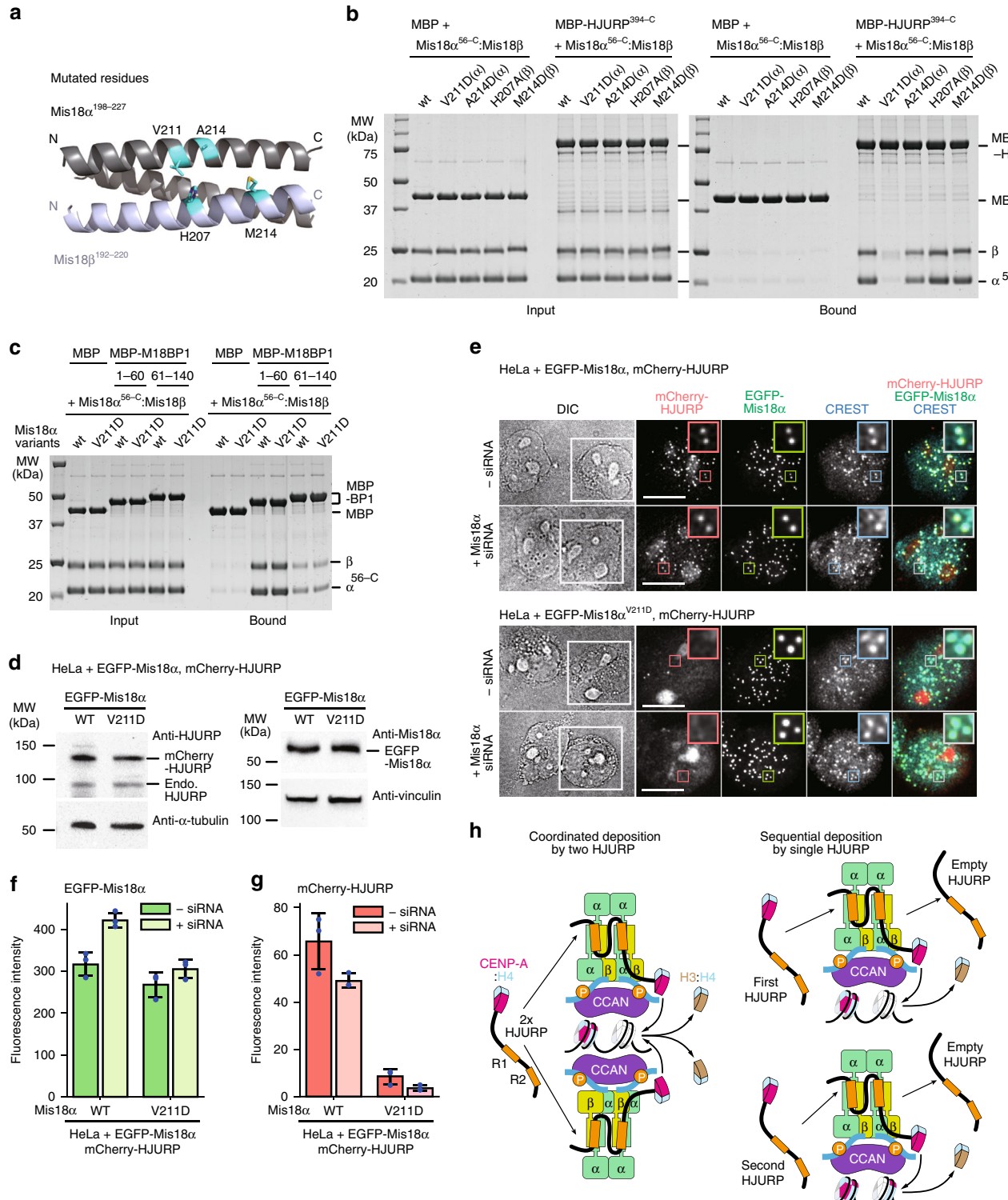

**Fig. 8** Mis18[V211D] is a separation-of-function mutant with impaired HJURP binding but not M18BP1 binding. **a** Mutated residues of Mis18α and Mis18β are indicated on the 3D model. **b**, **c** SDS-PAGE results of the amylose-resin pull-down assays for testing the interaction of the mutants of Mis18α and Mis18β with HJURP[394–C], M18BP1[1–60], and M18BP1[61–140]. **d** Western blotting analysis confirming transgene expression. Endo. endogenous. **e** Representative images showing the fluorescence of mCherry-HJURP and EGFP-Mis18α variants in fixed HeLa cells treated as described in Fig. 2a but without the blocking and labeling steps of SNAP-CENP-A. Mis18α siRNA was used instead of HJURP siRNA. White scale bars indicate 10 μm. **f**, **g** Quantification of the centromere fluorescence intensity of mCherry-HJURP and EGFP-Mis18α variants. The bar graphs represent mean values from three replicate experiments (blue dots indicate the mean values from each experiment). Error bars indicate standard deviations. **h** Two models of HJURP recruitment to the Mis18 receptor and the deposition of CENP-A:H4 dimers. Source data are provided as a Source Data file

4 °C for cleavage of the 6His-tag and MBP-tag and were purified using the Superdex 200 10/300 GL SEC column with buffer HST300.

The Bpa-incorporated or AbK-incorporated MBP-HJURP variants were expressed in *E. coli* cells of BL21(DE3) strain (Agilent Technologies, #200131). Plasmids carrying MBP-HJURP genes with TAG codon at the specific positions were used for transformation of the *E. coli* cells together with pEVOL-pBpF or pEVOL-ABK. The cells were cultured in 2 × YT media supplemented with ampicillin, chloramphenicol and 0.2% arabinose at 37 °C. Protein expression was induced by adding IPTG to the final concentration of 0.2 mM and the unnatural amino-acids (Bpa or AbK) to the final concentration of 1 mM when $OD_{600}$ of the culture reached 0.6. The culture was further incubated at 25 °C for 16 h. Purification of the Bpa-incorporated and AbK-incorporated MBP-HJURP variants was performed in the same way as normal MBP-HJURP variants described above.

**Amylose-resin pull-down assay.** Proteins were diluted to 5 μM in 40 μL binding buffer A (30 mM HEPES pH 7.5, 300 mM sodium chloride, 1 mM TCEP, 0.01% Tween-20) or binding buffer B (15 mM HEPES pH 7.5, 150 mM sodium chloride, 1 mM TCEP, 0.01% Tween-20), and mixed with 20 μL amylose resin (NEB) equilibrated with the binding buffer. One-third of this mixture was taken as input fraction and the rest two-thirds were incubated at 4 °C for 30 min. Amylose-bound proteins were separated from unbound fraction by spinning down the amylose resin and washing the resin with 500 μL binding buffer four times. The input and bound fractions were analyzed by Tricine–SDS-PAGE. The gels were stained with Coomassie brilliant blue (CBB).

**Analytical size-exclusion chromatography (SEC).** SEC experiments were performed on a calibrated Superose 6 increase 5/150 GL column (GE Healthcare). Purified protein samples were mixed at 10 μM (except Mis18α at 20 μM) and analyzed under isocratic condition at 4 °C in buffer containing 15 mM HEPES pH 7.5, 150 mM sodium chloride and 0.5 mM TCEP at a flow rate of 0.1 mL min$^{-1}$. Fractions were collected and analyzed by Tricine–SDS-PAGE and the gels were stained with CBB.

**Protein fluorescence labeling using sortase.** MBP-M18BP1$^{1-228}$-LPETGG was labeled with GGGGK peptides with a C-terminally conjugated fluorescein amidite (FAM) (Genscript) using purified Sortase 5 M mutant[79]. Labeling was performed for ~16 h at 4 °C by incubation of 100 μM MBP-M18BP1$^{1-228}$-LPETGG with 1 mM GGGGK-FAM peptides and 10 μM Sortase 5 M in the reaction buffer containing 50 mM HEPES pH 7.5, 500 mM NaCl, 1 mM TCEP and 10 mM CaCl$_2$. His-tagged Sortase was removed by incubation of the reaction mix with Ni resin (Roche). Excess of GGGGK-FAM peptides were removed by gel filtration.

**Analytical ultracentrifugation (AUC).** Sedimentation velocity AUC was performed at 42,000 rpm at 20 °C in a Beckman XL-A ultracentrifuge. Purified protein samples were diluted in buffer HST300 and loaded into standard double-sector centerpieces. The cells were scanned at 280 nm for unlabeled samples or 488 nm for FAM-labeled samples. More than 300 scans were recorded for each sample. Data were analyzed using the program SEDFIT[80] with the model of continuous $c(s)$ distribution. The partial specific volumes of the proteins, buffer density, and buffer viscosity were estimated using the program SEDNTERP. Figures were generated using the program GUSSI.

**UV-cross-linking and cross-link identification.** The Bpa-incorporated or AbK-incorporated MBP-HJURP variants were diluted in 200 μL buffer HST300 at 3 μM and were incubated with Mis18α$^{56-C}$:Mis18β 1.5 μM (calculated as hexamer). LED UV light of 365 nm (Nichia, NCSU276A) was used to irradiate the protein samples for 30 min on ice to activate the cross-linking reaction. The protein solutions were incubated with 10 μL Ni resin (Roche), and MBP-HJURP variants carrying 8His-tags at the C-termini bound on Ni resin together with Mis18α$^{56-C}$:Mis18β. Unbound Mis18α$^{56-C}$:Mis18β was removed by washing the resin with 500 μL buffer HST300 for three times. Uncross-linked Mis18α$^{56-C}$ and Mis18β were removed by washing the resin twice with 100 μL 8 M urea containing 10 mM HEPES pH 7.5 and 1 mM TCEP. Samples were collected at each step and analyzed using Tricine-SDS-PAGE. Cross-linked protein samples on Ni resin were incubated with 8 M urea containing 1 mM DTT at 25 °C for 30 min. Chloroacetamide was added to the solution to the final concentration of 5.5 mM for alkylation. LysC and trypsin were used to digest the samples at 25 °C for ~16 h, and the digestion was stopped by adding trifluoroacetic acid (TFA) to the final concentration of 0.2%. Peptides were purified using Sep-Pak tC18 cartridges (50 mg, Waters), eluted in water containing 60% acetonitrile and 0.1% TFA, and dried in tubes completely.

LC-MS/MS analysis was performed as previously reported[81] using an Ultimate 3000 RSLC nano system and a Q-Exactive Plus mass spectrometer (Thermo Fisher Scientific). Peptides were dissolved in water containing 0.1% TFA and were separated on the Ultimate 3000 RSLC nano system (precolumn: C18, Acclaim PepMap, 300 μm × 5 mm, 5 μm, 100 Å, separation column: C18, Acclaim PepMap, 75 μm × 500 mm, 2 μm, 100 Å, Thermo Fisher Scientific). After loading the sample on the precolumn, a multistep gradient from 5–40% B (90 min), 40–60% B (5 min), and 60–95% B (5 min) was used with a flow rate of 300 nL min$^{-1}$; solvent A: water + 0.1% formic acid; solvent B: acetonitrile + 0.1% formic acid. Data were acquired

using the Q-Exactive Plus mass spectrometer in data-dependent MS/MS mode. For full scan MS, we used mass range of $m/z$ 300–1800, resolution of R = 140000 at $m/z$ 200, one microscan using an automated gain control (AGC) target of 3e6 and a maximum injection time (IT) of 50 ms. Then, we acquired up to 10 HCD MS/MS scans of the most intense at least doubly charged ions (resolution 17500, AGC target 1e5, IT 100 ms, isolation window 4.0 m/z, normalized collision energy 25.0, intensity threshold 2e4, dynamic exclusion 20.0 s). All spectra were recorded in profile mode.

Raw data from the Q-Exactive Plus mass spectrometer were converted to Mascot generic files (MGF) format with program msConvert GUI from ProteoWizard Toolkit version 3[82]. Program StavroX version 3.6.6.6[83] was used for cross-link identification. MS data in MGF format and the protein sequences in FASTA format were loaded on the program and the MS spectra matching cross-linked peptides were searched. In the settings of StavroX, the precursor precision and the fragment ion precision were changed to 10.0 and 20.0 ppm, respectively, from the default values. For the analysis of AbK-cross-linked samples, a new entry of AbK with the chemical composition of $C_{11}H_{18}N_4O_2$ was added to the settings section of amino-acids in StavroX. And a new entry of AbK was appended to the cross-linker settings section with the following information: name, ABK; composition, $-N_2$; maximum distance, 30 Å; site1, z; site2, {ABCDEFGHIKLMPQRSTVWY}, where '{' and '}' are the N- and C-termini, respectively, and the letters indicate single amino-acids, with B being alkylated Cys and z being AbK. StavroX estimates the false discovery rate (FDR) by comparison of the distribution of the cross-link candidates found using provided protein sequences and the distribution of the candidates found from decoy search using shuffled sequences, and 5% FDR was used as the cutoff to exclude the candidates with lower StavroX scores. The results of cross-link data were exported in comma-separated values (CSV) format. The cross-link candidates with mass deviation outside the range of 10 ppm (Bpa dataset, −3 to 7 ppm; AbK dataset, −4 to 6 ppm) were excluded (Supplementary Data 2). The cross-link maps of the intermolecular cross-links among HJURP, Mis18α and Mis18β were drawn using Adobe Illustrator software.

**Generation of stable HeLa cell lines.** All HeLa cell lines used in this study (Supplementary Data 1) were generated from a Flp-In T-REx HeLa cell line created by Stephen Taylor and colleagues[61]. The Flp-In T-REx HeLa cells were transfected with the pcDNA5/FRT/TO derived plasmids and the pOG44 plasmid (Thermo Fisher Scientific) according to the manufacture's protocol. Stable cell lines were generated by isolation of single colonies, which were viable in the DMEM medium containing 10% tetracycline-free FBS, 2 mM l-glutamine, 250 μg mL$^{-1}$ hygromycin and 4 μg mL$^{-1}$ blasticidin. Protein expression was verified using western blotting (uncropped blot images are in a Source Data file) and the following antibodies were used: anti-HJURP antibody (Abcam, ab100800; dilution, 1:500), anti-Mis18α antibody (Thermo Fisher Scientific, PA5-53771; dilution, 1:500), anti-SNAP-tag antibody (NEB, P9310; dilution, 1:1,000), anti-α-tubulin antibody (Sigma, T9026; dilution, 1:8,000), anti-vinculin antibody (Sigma, V9131; dilution, 1:10,000), anti-mouse HRP-conjugated antibody (GE, NXA931-1ML; dilution, 1:10,000), anti-rabbit HRP-conjugated antibody (GE, NA934-1ML; dilution, 1:10,000).

**CENP-A deposition experiment.** HeLa cells were treated with 10 nM HJURP Stealth RNAi siRNA targeting the 3′-untranslated region of HJURP gene (Thermo Fisher Scientific) or 10 nM Mis18α siRNA[12] using Lipofectamine RNAiMAX according to the manufacture's protocol of reverse transfection and were incubated in DMEM supplemented with 10% tetracycline-free FBS, 2 mM l-glutamine, 100 U mL$^{-1}$ Penicillin and 0.1 mg mL$^{-1}$ Streptomycin (PAN-Biotech) at 37 °C in the presence of 5% CO$_2$ for 48 h. Doxycycline (Sigma) was added to the culture at a concentration of 50 ng mL$^{-1}$ to induce protein expression and was kept in the media until the fixation of cells. Thymidine (1 mM final concentration) was used to arrest cells at S/G1 transition phase. When cells were released from thymidine, existing SNAP-CENP-A proteins were blocked using SNAP-Cell Block or SNAP-Cell TMR-star (NEB) according to the manufacture's protocol. STLC (5 mM final concentration) was used to arrest cells in prometaphase. The cells arrested in prometaphase were separated from other cells by mitotic-shake-off, released from STLC by extensive wash with the media and placed in wells of 24-well plates containing poly-lysine coated coverslips. Three hours later, the cells in early G1 phase attached on the coverslips and were treated with SNAP-Cell 647-SiR (NEB) to label newly synthesized SNAP-CENP-A according to the manufacture's protocol. Then the cells were incubated with pre-extraction buffer (100 mM sodium chloride, 300 mM sucrose, 3 mM MgCl$_2$, 10 mM PIPES pH 6.8, 0.1% Triton X-100) on ice for 2 min and fixed with PBS containing 4% paraformaldehyde at room temperature for 20 min.

The fixed cells were permeabilized with PBS-T (PBS buffer containing 0.1% Triton X-100) for 10 min and incubated with PBS-T containing 4% BSA for 40 min. CREST/anti-centromere antibody (Antibodies, Inc.; dilution, 1:200) and anti-human DyLight 405–conjugated secondary antibody (Jackson ImmunoResearch; dilution, 1:200) were used for immunostaining of the centromeres. After washing and drying, the coverslips were mounted with Mowiol mounting media (EMD Millipore) on glass slides and imaged using a ×60 oil immersion objective lens on a DeltaVision deconvolution microscope. Quantification of centromere signals was performed using the software Fiji[84] with a script for semiautomated processing. Briefly, average projections were made from z-stacks of recorded images. Centromere spots were chosen based on the

parameters of shape, size, and intensity using the images of the reference channel obtained with CREST-staining, and their positions were recorded. In the images of the data channels, the mean intensity value of adjacent pixels of a centromere spot was subtracted as background intensity from the mean intensity value of the centromere spot. Statistical analysis of the quantified intensity was performed using Microsoft Excel and the plots were generated using Matplotlib libraries.

**Structure modeling of the Mis18 C-terminal helix-bundle**. The structure model of the trimeric C-terminal helix-bundle of Mis18α and Mis18β was made using CCBuilder 2.0[67]. Amino-acid sequences of Mis18α Arg198–Phe227 and Mis18β Leu192–Val220 were used to generate a trimeric coiled-coil bundle. Chain 1 and chain 2 were assigned to Mis18α 198–227, and chain 3 to Mis18β 193–220. Radius was set to 6 Å, pitch to 200 Å, interface angle to 24°. The register of Mis18α Arg198 was set to "a", and the register of Mis18β Leu192 to "d".

**Reporting summary**. Further information on research design is available in the Nature Research Reporting Summary linked to this article.

## Data availability

The mass spectrometry data have been deposited to the ProteomeXchange Consortium via the PRIDE partner repository with the dataset identifier PXD013339. All other relevant data supporting the key findings of this study are available within the article and its Supplementary Information files or from the corresponding authors upon reasonable request. The source data underlying Figs. 2, 3, 8 and Supplementary Figs. 3, 5, 6 are provided in a Source Data file. A reporting summary for this Article is available as a Supplementary Information file.

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

## Acknowledgements

D.P. gratefully acknowledges support from the Alexander von Humboldt Foundation through Humboldt Research Fellowships. A.M. gratefully acknowledges funding by the Max Planck Society, the European Research Council (ERC) Advanced Investigator Grant RECEPIANCE (proposal number n° 669686), and the DFG's Collaborative Research Centre (CRC) 1093. We thank Heinz Neumann for sharing reagents and for help with the setup of the amber-codon suppression experiments. We thank all members of the Musacchio laboratory for helpful discussions and comments, and Franziska Müller and Andreas Brockmeyer for support of MS analysis.

## Author contributions

D.P. generated plasmids. D.P., A.T., and D.B. purified proteins and performed biochemical assays. D.P., K.W., and A.T. created HeLa cell lines and performed CENP-A deposition assays and data analysis. D.P. performed AUC experiments and UV-crosslinking experiments. N.K. synthesized AbK. A.M. supervised and administered the research team. D.P. and A.M. wrote the manuscript with contributions from all authors.

## Additional information

**Competing interests:** The authors declare no competing interests.

