## [Peer Review File · Nature Communications]

Reviewers' comments:

Reviewer #1 (Remarks to the Author):

This manuscript investigates the precise molecular interactions occurring between the key human centromere assembly components HJURP, Mis18- α and - β (=Mis18core) and M18BP1. For this the authors utilise biochemical approaches (in vitro pulldown, sedimentation velocity, size exclusion chromatography), followed by validation using functional centromere localisation and assembly assays in cultured cells.

The study makes a number of novel findings with respect to the functional domains of HJURP required for centromere recruitment and assembly. Surprisingly, many of these observations do not agree with previous published reports from Zasadzinska et al. (EMBO J), Nardi et al. (Mol Cell) and Wang et al. (JBC). In my opinion, the novel in vitro and in vivo results presented here (1-4 highlighted below) are robust and convincing. Therefore these new results are sure to be of interest to the centromere field. Moreover, the findings of this study potentially alter the current view of the mechanism that limits CENP-A assembly at centromeres (i.e. active dissociation of the Mis18 complex by HJURP), leading the proposition of a modified version of the model (i.e. exhaustion of the 'H3' reaction substrate), which is of general interest to the wider community. However, given that several key observations do not agree with previous findings, I believe that the authors need to address and broaden their discussion (in either the results or discussion sections) to explain why different results have been obtained between this and existing studies. For example, could the observed differences be due to the size of the fragments used or the contribution of amino acids outside of the R1 and R2 domains?

The authors make the following major claims (1-4 below). Results 1-3 disagree with previous observations for which further explanation should be provided:

1. Both R1 (=HCTD1) and R2 (=HCTD2) domains of HJURP bind the Mis18 core complex by in vitro pulldown assays and are required for HJURP recruitment (shown by either mutation or deletion) to centromeres in G1 phase and new CENP-A deposition in vivo. This differs from a previous study showing that a C terminal fragment of HJURP containing only the R1 domain was required for HJURP recruitment to centromeres, whereas a C terminal fragment harbouring only the R2 domain was required for HJURP dimerization and CENP-A deposition (Zasadzinska et al. 2013).
2. The authors report that the HJURP R2 domain is a monomer (demonstrated by sedimentation velocity experiments). This differs from previous studies which showed that a HJURP fragment containing the R2 domain behaves as a dimer both in vitro and in vivo (Zasadzinska et al. 2013, Wang

et al. 2014). This new result has implications for the mechanism by which HJURP is recruited to centromeres and for the mechanism by which two CENP-A-H4 dimers are deposited.

3. Mis18- α and - β , M18BP1 and HJURP form a single complex in size exclusion chromatography and sedimentation velocity experiments. The authors find no evidence for the dissociation of the Mis18 core by HJURP as previously proposed by Nardi et al. 2016. This result has further implications for the mechanism by which CENP-A deposition at centromeres is completed.

4. Lastly, the authors report novel data on regions of the C terminal helices of Mis18- α and - β that are required to bind HJURP and M18BP1. The authors introduce photo-activatable cross-linkers into R1 and R2 by amber codon suppression to isolate an interesting Mis18- α V211D separation of function mutant that is required for HJURP recruitment, but can still bind M18BP1. This powerful approach will prove useful to further dissect and elucidate the initial and concluding steps of the assembly process.

Other (minor) comments:

-Figure 2: The 4K2 mutant appears to show some/reduced centromere localisation. Is this the case (although this is not reflected in the quantitation)?

-References are inserted in an inconsistent format in many places. Also, reference 1 appears to be incorrect?

-TYPOs: Please fix

-p13 His207Mis18 should be His207Mis18 β

-p18 Chemical 'synthesis'

-Statistical analysis performed are clearly stated in the figure legends and methods are provided in sufficient detail to enable reproducibility of these experiments.

Reviewer #2 (Remarks to the Author):

Pan et al. report on a detailed study that investigates the interaction of the CENP-A associated chaperone, HJURP, with the Mis18 α -Mis18 β -M18BP1 complex. The authors characterize the HJURP domains responsible for binding, derive the stoichiometry of the four-subunit complex, and identify interaction sites between HJURP and Mis18a/b using photo-cross-linking of non-native amino acids. Given the important role of the system in chromosome segregation, the results

provided in this work help to understand structural and mechanistic details of the interaction between the complex members.

I have been asked to comment on the cross-linking related aspects of the manuscript specifically. The authors used non-native, photo-reactive amino acids introduced to the R1 and R2 domains of HJURP to study possible interaction regions on Mis18alpha and beta. Cross-linked residues were identified by mass spectrometry according to established procedures. The cross-linking experiments are described in sufficient detail, and the results are appropriately discussed. The putative binding regions on Mis18a/b derived from the cross-linking experiments are subsequently used to generate point mutations and study their impact on the affinity to HJURP.

I have only one minor comment related to the cross-linking experiments: In the present version of the manuscript, the cross-linking results are only summarized as residue-residue contacts in Figure 6. Although the original data has been deposited to PRIDE (which is greatly appreciated) and it would therefore be possible to retrieve this information indirectly, I would still consider it meaningful to add a separate table to the supporting information, where more information could be provided about the individual identifications (peptide sequences, scores etc.).

Additional minor comment:

On page 15, some references seem to be inappropriately cited by author/year, not by reference number.

Reviewed by Alexander Leitner, ETH Zurich, Switzerland

Reviewer #3 (Remarks to the Author):

Centromeres are specialised chromosomal regions act as a platform for the assembly of the kinetochore, a multi subunit protein assembly which physically connects chromosomes to the

spindle microtubules to achieve chromosome segregation during cell division. Identity of the centromeric chromatin is defined by the enrichment of cenp-A (a histone H3-variant) containing nucleosomes. This epigenetic mark (cenp-A containing nucleosomes) is diluted during DNA-replication due to the distribution of existing cenp-A to the newly synthesised DNA. Hence, to maintain the centromere identity cenp-A needs to be actively deposited to the right levels at centromeres. hJurp is a cenp-A specific chaperone responsible for the deposition of cenp-A at centromeres. While we know the players responsible for the timely centromere targeting of hJurp, the mis18 complex (mis18c; consisting of mis18a/mis18b/mis18BP1) and cenp-C, precise molecular mechanisms controlling the mis18 complex assembly, its centromere association and subsequent centromere targeting of hJurp are just emerging. In this manuscript, Pan et al aims to dissect the intermolecular interactions critical for the centromere targeting of hJurp by the mis18 complex using a combination of biochemical, biophysical and cell based approaches.

The authors show that a C-terminal fragment of hJurp containing HCTD1 (R1) and HCTD2 (R2) (sequence related C-terminal domains of hJurp) does not form a dimer as previously reported by the Foltz lab (Nardi et al, Mol Cell, 2016) and exists as a monomer and one copy of hJurp interacts with a hetero hexameric Mis18 complex (4 mis18a, 2 mis18b and 2 mis18BP1) to form a heptameric assembly. They also show that this heptameric mis18c-hJurp assembly is stable unlike previously suggested by the Nardi et al, that the mis18c disassembles into a smaller oligomeric form that cannot interact with hJurp. Using elegant biochemistry, the authors show that hJurp HCTD1 and HCTD2 can bind mis18c independently. Using SNAP-cenp-A cell lines they show that though redundant both HCTD1 and HCTD2 are required for correct function – not the identity, but dosage is important. Finally, using photoactivatable crosslinking MS they identify residues crucial for hJurp binding (mis18a V211) and demonstrate that mutating this residue while does not affect centromere association of the mis18c disrupts centromere targeting of hJurp by the mis18c.

Overall, this work using high quality biochemical and biophysical data reinvestigates the model proposed by the Foltz lab on how mis18c interacts with hJurp. Most conclusions reported here while contradict the previous study by the Foltz lab (Nardi et al., 2016) provide additional new insights on how the mis18c interacts with hJurp and identified a separation-of-function mis18a mutant that selectively perturbs hJurp binding. These are important observations and will be of interest to the centromere field. However, addressing the following concerns might strengthen the conclusions.

1. It is surprising that the entire biochemical characterisation of mis18c-hJurp interaction is carried out using MBP-tagged hJurp. Although the authors show that MBP shows only a very weak binding to mis18c in their affinity-pull down assay, one cannot rule out the possible influence of having such a bulky tag on artificially stabilizing/destabilising the structural dynamics of hJurp which might eventually have an effect on mis18c binding. I think it is important to characterise these interactions using untagged hJurp. If MBP is absolutely essential for solubilizing hJurp, the authors could try to cleave the tag after reconstituting MBP-hJurp-mis18c complex and show that the complex is stable and oligomeric structure is not affected or/and use a relatively smaller affinity tag.

2. Cenp-A deposition by hJurp containing R1-R1 is slightly more effective than R2-R2. Calorimetric binding studies with hJurp R1 – mis18c; hJurp R2 – mis18c; hJurp R1-R1-mis18c; hJurp R2-R2-mis18c and hJurp R1-R2-mis18c might explain the functional significance of R1-R2 dosage in cells.

3. Why deleting R1 and R2 results in dominant negative effect on cenp-A deposition needs to be discussed in detail, particularly as the authors show that hJurp do not form a dimer in vitro. It is important to analyse the full length hJurp either untagged or MBP-hJurp in AUC to see if hJurp has any weak tendency to oligomerise. Authors have used MBP hJurp 1-748 in an affinity pull down assay, based on this I assume that this version can be purified.

4. In Figure 1 D, MBP hJurp 1-748 appears to interact with mis18a/b weakly as compared to hJurp 394-748. This needs to be addressed with additional experiments and discussed appropriately.

5. In the absence of crystal structure, the authors have modelled the triple helical structure made of 2mis18a and 1mis18b using CCBUILDER 2.0. It is important to explicitly show different models (with different orientations, parallel or anti parallel) obtained from CCBUILDER 2.0 and by mapping intermolecular/intramolecular contacts seen between mis18a/b c-terminal helices, demonstrate that the chosen model is the likely to be the right one. It will also be important to validate this model in vitro and in cell using crosslinking guided mis18a or mis18b mutants.

Reviewer #1 (Remarks to the Author):

This manuscript investigates the precise molecular interactions occurring between the key human centromere assembly components HJURP, Mis18- α and - β (=Mis18core) and M18BP1. For this the authors utilise biochemical approaches (in vitro pulldown, sedimentation velocity, size exclusion chromatography), followed by validation using functional centromere localisation and assembly assays in cultured cells.

The study makes a number of novel findings with respect to the functional domains of HJURP required for centromere recruitment and assembly. Surprisingly, many of these observations do not agree with previous published reports from Zasadzinska et al. (EMBO J), Nardi et al. (Mol Cell) and Wang et al. (JBC). In my opinion, the novel in vitro and in vivo results presented here (1-4 highlighted below) are robust and convincing. Therefore these new results are sure to be of interest to the centromere field. Moreover, the findings of this study potentially alter the current view of the mechanism that limits CENP-A assembly at centromeres (i.e. active dissociation of the Mis18 complex by HJURP), leading the proposition of a modified version of the model (i.e. exhaustion of the 'H3' reaction substrate), which is of general interest to the wider community.

We thank the reviewer for a positive evaluation of our work.

However, given that several key observations do not agree with previous findings, I believe that the authors need to address and broaden their discussion (in either the results or discussion sections) to explain why different results have been obtained between this and existing studies. For example, could the observed differences be due to the size of the fragments used or the contribution of amino acids outside of the R1 and R2 domains?

Please see our answer to this specific question below.

The authors make the following major claims (1-4 below). Results 1-3 disagree with previous observations for which further explanation should be provided:

1. Both R1 (=HCTD1) and R2 (=HCTD2) domains of HJURP bind the Mis18 core complex by in vitro pulldown assays and are required for HJURP recruitment (shown by either mutation or deletion) to centromeres in G1 phase and new CENP-A deposition in vivo. This differs from a previous study showing that a C terminal fragment of HJURP containing only the R1 domain was required for HJURP recruitment to centromeres, whereas a C terminal fragment harbouring only the R2 domain was required for HJURP dimerization and CENP-A deposition (Zasadzinska et al. 2013).
2. The authors report that the HJURP R2 domain is a monomer (demonstrated by sedimentation velocity experiments). This differs from previous studies which showed that a HJURP fragment containing the R2 domain behaves as a dimer both in vitro and in vivo (Zasadzinska et al. 2013, Wang et al. 2014). This new result has implications for the mechanism by which HJURP is recruited to centromeres and for the mechanism by which two CENP-A-H4 dimers are deposited.
3. Mis18- α and - β , M18BP1 and HJURP form a single complex in size exclusion chromatography and sedimentation velocity experiments. The authors find no evidence for

the dissociation of the Mis18 core by HJURP as previously proposed by Nardi et al. 2016. This result has further implications for the mechanism by which CENP-A deposition at centromeres is completed.

4. Lastly, the authors report novel data on regions of the C terminal helices of Mis18- α and - β that are required to bind HJURP and M18BP1. The authors introduce photo-activatable cross-linkers into R1 and R2 by amber codon suppression to isolate an interesting Mis18- α V211D separation of function mutant that is required for HJURP recruitment, but can still bind M18BP1. This powerful approach will prove useful to further dissect and elucidate the initial and concluding steps of the assembly process.

This is a fair summary of our main conclusions. We can only agree with the reviewer that these data diverge from those reported previously by the Foltz laboratory. A fundamental premise is that we have worked under the highest possible standards of biochemical and biophysical investigations, as we feel the reviewer has recognized. Often, binding artefacts can occur if recombinant proteins are imperfectly folded and unstable. In the absence of direct formal evidence of this, however, we would find it awkward to allege that others have failed to meet the required standards, even if we consider this is a plausible explanation. In general, we feel that the manuscript refers to possible sources of discrepancies, whenever these have a recognizable explanation. In particular, we write:

1) “Binding of R2 of HJURP to the Mis18^{core} complex contradicts a previous report⁸. Because a shorter R2 construct (HJURP⁵⁵⁵⁻⁷⁴⁸ instead of HJURP⁵⁴¹⁻⁷⁴⁸ used here) had been used in the previous study⁸, we compared the Mis18^{core} binding proficiency of HJURP⁵⁵⁵⁻⁷⁴⁸ and HJURP⁵⁴¹⁻⁷⁴⁸. HJURP⁵⁴¹⁻⁷⁴⁸ bound Mis18^{core} markedly better than MBP-HJURP⁵⁵⁵⁻⁷⁴⁸ (Supplementary Fig. 2A), **likely explaining the different outcomes of the binding experiments**” (page 6). Here, we refer to a difference in construct boundaries that might have misled Nardi et al.

2) “R2 has been proposed to promote and be sufficient for HJURP dimerization³³. To verify this claim, we applied sedimentation velocity analytical ultracentrifugation (AUC), **a method of choice for accurate determination of molecular mass**” (page 6). Please see comments to point 3.

3) “Another important conclusion of our study is that binding to HJURP394–C does not change the oligomerization state of the Mis18core complex as previously proposed⁸. Although other regions of HJURP might be responsible for Mis18core dissociation in living cells, the fact that HJURP1-393 does not bind Mis18core conflicts with this idea. Furthermore, Nardi et al. (2016)⁸ observed dissociation of Mis18core by HJURP in vitro when using a C-terminal segment of HJURP with similar span to the one we have used. While we cannot pinpoint the reasons for these differences, we note that our conclusion that only one HJURP394–C binds to the Mis18 octamer and that the complex does not dissociate were obtained with **analytical ultracentrifugation, the gold standard for this purpose**” (page 15). In 2 and 3, we refer indirectly to the fact that Nardi *et al.* resorted to less accurate analytical methods such as density gradient centrifugation.

4) “A significant technical difference is that we used stable transgene integration at a single chromosomal site, **whereas Zasadzinska et al. (2013) operated under condition of transient-transfection of a pIC113 plasmid containing a constitutive CMV promoter, and thus likely under conditions of over-expression**” (Page 15). Here again we point to differences in expression levels as the source of the problem.

Other (minor) comments:

-Figure 2: The 4K2 mutant appears to show some/reduced centromere localisation. Is this the case (although this is not reflected in the quantitation)?

We agree with the reviewer that the HJURP 4K2 mutant showed very weak but recognizable localization to centromeres. However, the reduction of centromere signal of HJURP is very large and emerges very clearly when comparing images of HJURP WT and of the 4K2 mutant. Quantifications in Figure 2E further confirm that the centromere intensity of the 4K2 mutant is greatly reduced compared to HJURP WT, indicating that HJURP R2 is required for stable centromere localization. Furthermore, the 4K2 mutant did not support CENP-A deposition, as shown in Figure 2C and 2D, thus supporting the idea that the 4K2 mutation impaired both centromere localization and CENP-A loading function.

-References are inserted in an inconsistent format in many places. Also, reference 1 appears to be incorrect?

Thank you for pointing this out. We corrected and unified the style of citations.

-TYPOS: Please fix

-p13 His207Mis18 should be His207Mis18 β

-p18 Chemical 'synthesis'

We corrected these typos.

-Statistical analysis performed are clearly stated in the figure legends and methods are provided in sufficient detail to enable reproducibility of these experiments.

Reviewer #2 (Remarks to the Author):

Pan et al. report on a detailed study that investigates the interaction of the CENP-A associated chaperone, HJURP, with the Mis18 α -Mis18 β -M18BP1 complex. The authors characterize the HJURP domains responsible for binding, derive the stoichiometry of the four-subunit complex, and identify interaction sites between HJURP and Mis18a/b using photo-cross-linking of non-native amino acids. Given the important role of the system in chromosome segregation, the results provided in this work help to understand structural and mechanistic details of the interaction between the complex members.

We are grateful to Dr. Leitner for a positive evaluation of our work.

I have been asked to comment on the cross-linking related aspects of the manuscript specifically. The authors used non-native, photo-reactive amino acids introduced to the R1 and R2 domains of HJURP to study possible interaction regions on Mis18 α and β . Cross-linked residues were identified by mass spectrometry according to established

procedures. The cross-linking experiments are described in sufficient detail, and the results are appropriately discussed. The putative binding regions on Mis18a/b derived from the cross-linking experiments are subsequently used to generate point mutations and study their impact on the affinity to HJURP.

I have only one minor comment related to the cross-linking experiments: In the present version of the manuscript, the cross-linking results are only summarized as residue-residue contacts in Figure 6. Although the original data has been deposited to PRIDE (which is greatly appreciated) and it would therefore be possible to retrieve this information indirectly, I would still consider it meaningful to add a separate table to the supporting information, where more information could be provided about the individual identifications (peptide sequences, scores etc.).

We generated an Excel file as Supplementary Data 2 with tables showing additional details of the cross-links identified by StavroX.

Additional minor comment:

On page 15, some references seem to be inappropriately cited by author/year, not by reference number.

Thank you for pointing this out. We corrected and unified the style of citations.

Reviewed by Alexander Leitner, ETH Zurich, Switzerland

Reviewer #3 (Remarks to the Author):

Centromeres are specialised chromosomal regions act as a platform for the assembly of the kinetochore, a multi subunit protein assembly which physically connects chromosomes to the spindle microtubules to achieve chromosome segregation during cell division. Identity of the centromeric chromatin is defined by the enrichment of cenp-A (a histone H3-variant) containing nucleosomes. This epigenetic mark (cenp-A containing nucleosomes) is diluted during DNA-replication due to the distribution of existing cenp-A to the newly synthesised DNA. Hence, to maintain the centromere identity cenp-A needs to be actively deposited to the right levels at centromeres. hjurp is a cenp-A specific chaperone responsible for the deposition of cenp-A at centromeres. While we know the players responsible for the timely centromere targeting of hjurp, the mis18 complex (mis18c; consisting of mis18a/mis18b/mis18BP1) and cenp-C, precise molecular mechanisms controlling the mis18 complex assembly, its centromere association and subsequent centromere targeting of hjurp are just emerging. In this manuscript, Pan et al aims to dissect the intermolecular interactions critical for the centromere targeting of hjurp by the mis18 complex using a combination of biochemical, biophysical and cell based approaches.

The authors show that a C-terminal fragment of hjurp containing HCTD1 (R1) and HCTD2 (R2) (sequence related C-terminal domains of hjurp) does not form a dimer as previously reported by the Foltz lab (Nardi et al, Mol Cell, 2016) and exists as a monomer and one copy of hjurp interacts with a hetero hexameric Mis18 complex (4 mis18a, 2 mis18b and 2 mis18BP1) to form a heptameric assembly. They also show that this heptameric mis18c-hjurp

assembly is stable unlike previously suggested by the Nardi et al, that the mis18c disassembles into a smaller oligomeric form that cannot interact with hjurp. Using elegant biochemistry, the authors show that hjurp HCTD1 and HCTD2 can bind mis18c independently. Using SNAP-cenp-A cell lines they show that though redundant both HCTD1 and HCTD2 are required for correct function – not the identity, but dosage is important. Finally, using photoactivatable crosslinking MS they identify residues crucial for hjurp binding (mis18a V211) and demonstrate that mutating this residue while does not affect centromere association of the mis18c disrupts centromere targeting of hjurp by the misc18c.

Overall, this work using high quality biochemical and biophysical data reinvestigates the model proposed by the Foltz lab on how mis18c interacts with hjurp. Most conclusions reported here while contradict the previous study by the Foltz lab (Nardi et al., 2016) provide additional new insights on how the mis18c interacts with hjurp and identified a separation-of-function mis18a mutant that selectively perturbs hjurp binding. These are important observations and will be of interest to the centromere field. However, addressing the following concerns might strengthen the conclusions.

We are grateful to the reviewer for a positive evaluation of our work.

1. It is surprising that the entire biochemical characterisation of mis18c-hjurp interaction is carried out using MBP-tagged hjurp. Although the authors show that MBP shows only a very weak binding to mis18c in their affinity-pull down assay, one cannot rule out the possible influence of having such a bulky tag on artificially stabilizing/destabilising the structural dynamics of hjurp which might eventually have an effect on mis18c binding. I think it is important to characterise these interactions using untagged hjurp. If MBP is absolutely essential for solubilizing hjurp, the authors could try to cleave the tag after reconstituting MBP-hjurp-mis18c complex and show that the complex is stable and oligomeric structure is not affected or/and use a relatively smaller affinity tag.

We agree with the reviewer that the use of an MBP-tag might affect the structural organization and dynamics of HJURP and therefore followed his/her suggestion and carried additional experiments that allow us to exclude this possibility. Specifically, we performed AUC analysis of Mis18 α :Mis18 β :HJURP complex with and without cleavage of MBP-tag from HJURP as suggested by the reviewer. As shown in the new Figure 5 and Supplementary Fig. 6, HJURP394-C formed a stable complex with Mis18 α :Mis18 β even when the MBP-tag was removed. The removal of the MBP-tag from HJURP did not change the stoichiometry of Mis18 α :Mis18 β :HJURP complex.

2. Cenp-A deposition by hjurp containing R1-R1 is slightly more effective than R2-R2. Calorimetric binding studies with hjurp R1 – mis18c; hjurp R2 – mis18c; hjurp R1-R1-mis18c; hjurp R2-R2-mis18c and hjurp R1-R2-mis18c might explain the functional significance of R1-R2 dosage in cells.

The reviewer is correct that CENP-A deposition in absence of endogenous HJURP by HJURP containing R1-R1 may appear slightly more effective than R2-R2 (Figure 3B). However, an unpaired t-test using the mean values obtained from the triplicate experiments, included here, indicates that this difference is not statistically significant. The two-tailed P value was 0.17. We therefore removed the sentence “...(with R1-R1 appearing slightly more effective than R2-

R2)” from the text because we realized that it might mislead readers. The results of pull-down assays in Fig. 3D also showed no significant difference of Mis18 $\alpha\beta$ -binding among HJURP

Figure 3B

constructs R1-R2, R1-R1, and R2-R2. Based on this extensive evidence, we believe that the additional calorimetric binding studies suggested by the reviewer, which would be technically very demanding given the amounts of proteins required and the need to establish the assay, would not provide new critical information.

3. Why deleting R1 and R2 results in dominant negative effect on cenp-A

A deposition needs to be discussed in detail, particularly as the authors show that hjurp do not form a dimer in vitro. It is important to analyse the full length hjurp either untagged or MBP-hjurp in AUC to see if hjurp has any weak tendency to oligomerise. Authors have used MBP hjurp 1-748 in an affinity pull down assay, based on this I assume that this version can be purified.

As shown by Zasadzinska and colleagues (2013) and confirmed in our unpublished results, the R1-R2 fragment of HJURP is sufficient for KT localization. This means that most likely endogenous HJURP can reach the kinetochore unhindered when we express the mutant lacking R1 and R2 (regretfully we are unable to formally show that endogenous HJURP is correctly recruited to the kinetochore under these conditions due to lack of an antibody of adequate quality). Based on these observations and assumptions, we conclude that most likely there is no dimerization domain not only in the R1-R2 segment, but in the entire HJURP. If a dimerization domain existed outside of R1-R2, our constructs lacking R1 or R2 would dimerize with endogenous HJURP at the kinetochore and be recruited there, which is something we don't see. Thus, we have no reason to believe that the (partial) dominant negative effect is due to dimerization with endogenous HJURP. Rather, as we already explain in the manuscript, the simplest and most likely explanation for the dominant-negative effect of these mutants is that they bind to CENP-A via their intact N-terminal CENP-A binding domain, subtracting CENP-A from the endogenous protein and reducing the pool available for chromatin loading (we write “possibly because they compete with endogenous HJURP for CENP-A or other interaction partners” (Page 7).

4. In Figure 1 D, MBP hjurp 1-748 appears to interact with mis18a/b weakly as compared to hjurp 394-748. This needs to be addressed with additional experiments and discussed appropriately.

We suspect that this is an effect of the relatively low yields and less than ideal stability of the MBP-HJURP 1-748 construct, which results in a lower active fraction in the binding assays (an additional reason why we carrying out the experiment suggested under point 3 would be very demanding). We cannot exclude, as hinted to by the reviewer, that this conceals a regulatory step, but we feel that addressing this question would go beyond the scope of this already greatly data-rich manuscript.

5. In the absence of crystal structure, the authors have modelled the triple helical structure made of 2mis18a and 1mis18b using CCBUILDER 2.0. It is important to explicitly show different models (with different orientations, parallel or anti parallel) obtained from CCBUILDER 2.0 and by mapping intermolecular/intramolecular contacts seen between mis18a/b c-terminal helices, demonstrate that the chosen model is the likely to be the right one. It will also be important to validate this model in vitro and in cell using crosslinking guided mis18a or mis18b mutants.

We are slightly puzzled by the reviewer's comment, because we do validate the model in vitro and in cells, to a point that led us to isolate a separation of function Mis18 mutant (EGFP-Mis18 α^{V211D}) that fails to bind HJURP. Importantly, our goal was not to investigate whether

Figure 7 CH model
Residues are colored in the way presented on Fig. 7G, HJURP R2 panel.

the helices are parallel or anti-parallel, but rather to rationalize the crosslinking data to allow us to identify the most likely reciprocal organization of residues involved in HJURP binding through cross-linking. Our preference for one model over the other is that in case of an antiparallel orientation, the crosslinks are significantly more scattered on the structure than they are in the parallel orientation. This is shown in the neighboring scheme, which we present here on behalf

of the reviewer, where we present a model with the two helices of Mis18 α in anti-parallel orientation (instead of the parallel orientation shown in the manuscript).

REVIEWERS' COMMENTS:

Reviewer #3 (Remarks to the Author):

Authors have adequately addressed the concerns raised by this reviewer.

REVIEWERS' COMMENTS:

Reviewer #3 (Remarks to the Author):

Authors have adequately addressed the concerns raised by this reviewer.

We thank the reviewer for his/her support of our work